# A Distinct Microglial Cell Population Expressing Both CD86 and CD206 Constitutes a Dominant Type and Executes Phagocytosis in Two Mouse Models of Retinal Degeneration

**DOI:** 10.3390/ijms241814236

**Published:** 2023-09-18

**Authors:** Yan Zhang, Yong Soo Park, In-Beom Kim

**Affiliations:** 1Department of Anatomy, College of Medicine, The Catholic University of Korea, Seoul 06591, Republic of Korea; 00zhangyan00@naver.com (Y.Z.); yongsoopark88@gmail.com (Y.S.P.); 2Catholic Neuroscience Institute, College of Medicine, The Catholic University of Korea, Seoul 06591, Republic of Korea; 3Department of Biomedicine & Health Sciences, Graduate School, The Catholic University of Korea, Seoul 06591, Republic of Korea; 4Catholic Institute for Applied Anatomy, College of Medicine, The Catholic University of Korea, Seoul 06591, Republic of Korea

**Keywords:** microglia, CD86, CD206, phagocytosis, retinal degeneration

## Abstract

Microglial cells are the key regulators of inflammation during retinal degeneration (RD) and are conventionally classified as M1 or M2. However, whether the M1/M2 classification exactly reflects the functional classification of microglial cells in the retina remains debatable. We examined the spatiotemporal changes of microglial cells in the blue-LED and NaIO_3_-induced RD mice models using M1/M2 markers and functional genes. TUNEL assay was performed to detect photoreceptor cell death, and microglial cells were labeled with anti-IBA1, P2RY12, CD86, and CD206 antibodies. FACS was used to isolate microglial cells with anti-CD206 and CD86 antibodies, and qRT-PCR was performed to evaluate *Il-10*, *Il-6*, *Trem-2*, *Apoe*, and *Lyz2* expression. TUNEL-positive cells were detected in the outer nuclear layer (ONL) from 24 h to 72 h post-RD induction. At 24 h, P2RY12 was decreased and CD86 was increased, and CD86/CD206 double-labeled cells occupied the dominant population at 72 h. And CD86/CD206 double-labeled cells showed a significant increase in *Apoe*, *Trem2*, and *Lyz2* levels but not in those of *Il-6* and *Il-10*. Our results demonstrate that microglial cells in active RD cannot be classified as M1 or M2, and the majority of microglia express both CD86 and CD206, which are involved in phagocytosis rather than inflammation.

## 1. Introduction

Microglia are resident immune cells of the central nervous system (CNS), including the retina [1,2,3]. They regulate the CNS microenvironment, synaptic development, and structural remodeling in the normal state [4,5,6] and function as pro- or anti-inflammatory mediators and phagocytic cells under pathological conditions [7,8,9,10]. Thus, microglia have emerged as a therapeutic target for the treatment of various neurodegenerative diseases [7,11,12].

Microglial cells play an important role in retinal degeneration (RD), including age-related macular degeneration (AMD). AMD is the most common type of RD and leads to blindness due to irreversible photoreceptor and retinal pigment epithelium (RPE) degeneration [13,14]. Inflammation is considered an essential component of RD pathogenesis [15,16], and microglia/macrophage infiltration has been reported in AMD patients [17,18,19,20] and animal RD models [21,22,23]. However, the expected role of microglial cells in RD remains controversial. Microglial cells could mediate inflammation during RD by releasing pro-inflammatory cytokines, including interleukin-1β (IL-1β) and interleukin-6 (IL-6) [24,25,26,27]; on the other hand, it could mediate the anti-inflammatory process by releasing anti-inflammatory cytokines like interleukine-10 (IL-10) [28,29,30,31]. Conventionally, pro-inflammatory microglial cells are classified as M1 and are characterized by CD80, CD86, CD32, and CD11b expression. Anti-inflammatory and phagocytic microglial cells are defined by M2 and are characterized by CD206, arginase 1 (Arg 1), and resistin-like-α (FIZZ1) expression [32,33,34]. However, growing evidence suggests that microglial cells cannot be simply classified as M1 and M2 [35,36,37,38]. A subset of microglial cells modulates various neurodegenerative diseases [39,40], and recent progress in single-cell RNA sequencing offered a detailed functional classification of microglial cells in the RD retina [41,42], which is not included in the simple M1/M2 classification. In addition, discrimination of the infiltrated macrophage and innate microglia also emerged as an important issue in understanding the role of microglial cells during RD [40,41,42] because controlling the different origins of microglia has been expected as an important regulator for RD [43,44]. However, comprehensive information is still lacking about the morphological and functional characteristics of microglial cells with conventional M1/M2 or homeostatic markers, depending on the stage of RD. Because even single-RNA sequencing suggests new insight into microglial cells, M1/M2 markers still have value as a control mechanism of microglial cells. Thus, the expression of the M1 and M2 markers needs to be re-evaluated during RD for the right use of the conventional M1/M2 markers and isolation of the subtypes of microglial cells.

In this study, we examined the expression of representative M1 and M2 markers (CD86 and CD206) in microglial cells and their movement across retinal layers during RD progression. We then sorted microglial cells using anti-CD86 and CD206 antibodies and evaluated various functional genes using quantitative real-time polymerase chain reaction (qRT-PCR). Our findings suggest that the M1/M2 classification may not accurately reflect microglial cell function in the retina during RD. Instead, a dominant population of microglial cells in active RD may have a phagocytic function rather than modulating inflammation.

## 2. Results

### 2.1. Changes in Morphology and Distribution of Microglial Cells in Blue LED-Induced RD

First, we observed the distribution and morphology of microglial cells labeled with anti-IBA1 antibody to label pan-microglial cells during the progression of blue LED-induced RD using the TUNEL assay (Figure 1). In the RD retina, a few TUNEL-positive photoreceptors in the outer nuclear layer (ONL) started to appear at 12 h after RD, peaked at 72 h, and decreased at 120 h (Figure 1B–E). As a result, the retina, especially the ONL, became thinner; that is, 13–15 rows of photoreceptors in the ONL in normal controls became ~5 rows at 120 h after RD (Figure 1A–E). In the normal retina, horizontally ramified IBA1-labeled microglial cells were located in the outer plexiform layer (OPL) and inner plexiform layer (IPL) (Figure 1A,A’). Microglial cells had started to migrate to the ONL at 12 h after RD, and their horizontally ramified processes turned to the outer retina and elongated (Figure 1B,B’). Between 24 and 72 h after RD, microglial cells with enlarged cell bodies were mainly found in the ONL and subretinal space (SRS), and their vertically elongated processes mostly disappeared at 72 h (Figure 1C,D).

Changes in the number of microglial cells during RD according to the retinal layers are presented in Figure 1F–I. Throughout the whole retina, including the SRS, the total number of IBA1-labeled microglial cells started to increase at 12 h after RD induction, peaked at 72 h (*p* < 0.05), and decreased at 120 h (Figure 1F). In the ONL, microglial cells rapidly increased at 12 h after RD (*p* < 0.05), slightly decreased at 24 h, and peaked at 72 h (*p* < 0.05, Figure 1G). Unlike the ONL, microglial cells from the OPL to the ganglion cell layer (GCL) abruptly decreased at 12 h after RD (*p* < 0.05) and then gradually increased over time (Figure 1H). In the SRS, microglial cells were significantly increased at 72 h after RD but decreased at 120 h (*p* < 0.05, Figure 1I). These results suggest that the number of microglial cells increased during RD progression. Based on their morphology and distribution over time, they mainly migrated from the OPL and IPL, situated in the ONL, to SRS when photoreceptor cell death peaks and are then gradually re-distributed.

### 2.2. Innate Microglial Cell Response in Blue LED-Induced RD

Microglial cells with vertically elongated processes were observed in the ONL in the blue LED-induced RD retina at 12 h after RD (Figure 1B), which suggests early migration of the innate retinal microglial cells. To test whether these migrating microglial cells are innate and demonstrate their response during RD, we labeled them with an anti-P2RY12 antibody, an innate microglial cell marker [28,42], and an anti-IBA1 antibody because cells co-labeled with both antibodies represented innate microglial cells.

In the normal state, all microglial cells labeled with anti-IBA1 antibody within the retina exhibited P2RY12 immunoreactivity (Figure 2A). At both 12 and 24 h in the early period of RD, similar to the normal state, most microglial cells within the retina, including the ONL where migrating microglial cells were observed, were P2RY12-labeled, while approximately half of the microglial cells in SRS exhibited P2RY12 immunoreactivity (Figure 2B,C). Interestingly, at 72 h after RD, most microglial cells within the retina did not exhibit P2RY12 immunoreactivity (Figure 2D). In SRS, progressive loss of IBA1/P2RY12-co-labeled microglial cells was observed from 24 to 72 h after RD (Figure 2C,D). In the retina, at 120 h after RD, most microglial cells labeled in the ONL and SRS did not exhibit P2RY12 immunoreactivity, while co-labeled microglial cells positioned from the OPL to the GCL were often observed (Figure 2E).

Quantitatively, the number of IBA1/P2RY12-co-labeled microglial cells increased at 12 h post-RD, peaked at 24 h, and abruptly decreased at 72 h, when the IBA1-labeled microglial cell number was highest (Figure 2F). However, the proportion of IBA1/P2RY12-co-labeled microglial cells changed at different time points in each layer of the retina. The decrease in P2RY12 among the IBA1-labeled microglial cells was prominent between 24 and 72 h after RD in the ONL and OPL to GCL (ONL: 96.7% to 12.3%, OPL to GCL: 100% to 3.7%, *p* < 0.05). In SRS, the decrease in P2RY12 started at 12 h after RD (62.5%) and gradually decreased from 24 to 120 h after RD (24 h: 50.0%, 72 h: 36.0%, 120 h: 0%). Between 72 and 120 h after RD, only the OPL to GCL layers exhibited recovery of P2RY12 in IBA1-labeled microglial cells (3.7% to 15.6%).

These results suggest that retinal microglial cells migrating to the ONL and SRS in the early phase of RD are innate ones, and their characteristics appear to abruptly change because P2RY12, the representative innate microglia marker, was shifted from positive to negative at the peak time point of RD when maximal photoreceptor degeneration and reactive microglial cell infiltration occur.

### 2.3. Spatiotemporal Change of CD86 and CD206 Expression in Retinal Microglial Cells in Blue LED-Induced RD

Activated microglial cells have frequently been classified as M1 and M2 microglia responsible for pro-inflammation and anti-inflammation, respectively [32,33,34]. To elucidate whether M1 and M2 classification is applicable in RD, we examined the spatial and temporal distribution of microglial cells in blue LED-induced RD using triple-labeling experiments with anti-IBA1 antibodies and two representative M1 and M2 markers: anti-CD86 antibody [45,46] and anti-CD206 antibody [10,45], respectively.

In normal retinas, inactive IBA1-labeled microglial cells did not express CD86 and CD206 (Figure 3A). In the early phase (up to 24 h after RD), IBA1/CD86-double-labeled cells were observed in the ONL and SRS and were rarely observed in other retinal layers, while IBA1/CD206-double-labeled cells did not appear in the retina and SRS during this period (Figure 3B,C). In addition, a small number of IBA1/CD86/CD206-triple-labeled microglial cells were found in the SRS at 24 h after RD (Figure 3C). At 72 h after RD, when RD and microglial cell infiltration peak, IBA1/CD86/CD206-triple-labeled cells were most frequently observed in the ONL and SRS, while both IBA1/CD86-double-labeled and IBA1/CD206-double-labeled microglial cells were rarely observed (Figure 3D). The findings at 120 h after RD were similar to those at 72 h, except that the IBA1/CD206-double-labeled cells were a bit more frequently observed (Figure 3E).

Based on these findings, we performed a quantitative analysis (Figure 3F–I). Throughout the experimental period, IBA1-labeled microglial cells in the inner retina rarely expressed CD86 and/or CD206 (Figure 3H). However, IBA1/CD86-double-labeled cells corresponded to 45.4% of microglial cells at 12 h and 71.6% at 24 h after RD in the ONL (Figure 3G), and 82.2% at 12 h and 84.6% at 24 h after RD in the SRS (Figure 3I). At 72 h after RD, 84.7% and 81.7% of microglial cells in the ONL and SRS were CD86-labeled, respectively, and more than 90% of them expressed CD206. Thus, 77.6% and 80.0% of microglial cells in the ONL and SRS, respectively, were IBA1/CD86/CD206-triple-labeled cells, exhibiting an abrupt increase at 72 h after RD (*p* < 0.05). IBA1/CD86-double-labeled cells accounted for only 7.1% and 1.7% of microglial cells in the ONL and SRS groups, respectively, and IBA1/CD206-double-labeled cells accounted for 6.3% and 6.5% of microglial cells in the ONL and SRS groups, respectively (Figure 3G,I). At 120 h after RD, the number of microglial cells significantly decreased to about half of that found in the ONL and SRS at 72 h, but the IBA1/CD86/CD206-triple-labeled population was still dominant (ONL: 49.4%, SRS: 58.2%) (Figure 3G,I).

These results demonstrate that the conventional M1 and M2 markers, CD86 and CD206, exhibit complex expression patterns in the retinal microglia in blue LED-induced RD. This suggests that microglial cells in the retina could not be classified into just M1 and M2 phenotypes. Nevertheless, the expression pattern of CD86 and CD206 markers is critical to understand the dynamics of microglial cells, and co-expression of CD86 and CD206 indicates the dominant population of microglial cells during RD.

### 2.4. Characterization of IBA1/CD86/CD206-Triple-Labeled Microglial Population in Blue LED-Induced RD

To characterize IBA1/CD86/CD206-triple-labeled microglial cells in blue LED-induced RD, we first isolated them using FACS. An anti-CX3CR1 antibody, which is a common marker for whole microglial cells in the retina, similarly used as IBA1 [20,47], and an anti-CD206 antibody were used to sort the IBA1/CD86/CD206-triple-labeled population because 92.8% of the CD206-labeled microglial cells (vs. 91.5% of the CD86-labeled microglial cells) belonged to the IBA1/CD86/CD206-triple-labeled population that constituted the majority of the total retinal microglial cells at 72 h after RD (Figure 3D,G).

As shown in Figure 4A, microglial cells from the normal and 72 h post-RD retinas were sorted using FACS with anti-CX3CR1 antibody and anti-CD206 antibody. We divided the sorted cells into CD206^high^/CX3CR1 and CD206^low^/CX3CR1 microglial cells to compare the expression levels of the representative genes based on CD206 expression. The number of CD206^high^/CX3CR1 cells was significantly increased at 72 h (*p* < 0.05) in RD retinas compared with that in normal retinas (Figure 4B), and the difference in CD206 gene expression level was confirmed between CD206^high^/CX3CR1 and CD206^low^/CX3CR1 (*p* < 0.05, Figure 4C).

To understand the function of the CD206-labeled microglial cells corresponding to IBA1/CD86/CD206-triple-labeled cells at 72 h after RD, we examined the expression levels of genes that represent pro-inflammatory (*Il-6*), anti-inflammatory (*Il-10*), and phagocytosis and related metabolism (*Trem2*, *Lyz2*, and *Apoe*) genes. We measured the fold-change of each gene in CD206^high^/CX3CR1 cells compared with that observed in CD206^low^/CX3CR1 cells. Interestingly, *Il-6* and *Il-10* gene expression levels were not statistically different between CD206^high^/CX3CR1 cells and CD206^low^/CX3CR1 cells (*p* > 0.05, Figure 4D), whereas the expression of *Trem2*, *Lyz2*, and *Apoe*, defined as disease-associated microglial genes [48,49,50], were upregulated in the CD206^high^/CX3CR1 cells (*p* < 0.05, Figure 4D) compared with the CD206^low^/CX3CR1 cells. These results suggest that IBA1/CD86/CD206-triple-labeled cells are mainly involved in phagocytosis rather than retinal inflammation in RD. This assumption was supported by immuno-EM experiments performed with anti-CD206 antibodies. As shown in Figure 4E, we frequently observed CD206-labeled microglial cells that engulfed the cell body of the degenerating photoreceptors and contained many vacuoles. Moreover, their phagocytic character was further confirmed using triple-labeling experiments with anti-IBA1 antibody, anti-CD206 antibody, and anti-TREM2 antibody, which mediate target recognition and phagocytosis [51,52]. TREM2 was not expressed in microglial cells in the normal retina (Figure 4F), while it was labeled in IBA1/CD206-double-labeled microglial cells in the ONL and SRS at 72 h after RD (Figure 4G). Taken together, these results suggest that IBA1/CD86/CD206-triple-labeled microglial cells mainly have phagocytic functions in RD.

### 2.5. Microglial Cell Response and Character in the NaIO_3_-Induced RD

In blue LED-induced RD, innate retinal microglial cells migrated to the RD site in the early phase of RD. Furthermore, their characteristics changed, and their marker P2RY12 disappeared at the peak time point of RD. In addition, their molecular phenotype was mainly IBA1/CD86/CD206-immunopositive at this time. To test whether these microglial cell responses and characteristics usually appear during RD, we examined them in NaIO_3_-induced RD, a well-known RD model characterized by photoreceptor degeneration combined with rapid and massive retinal pigment epithelial (RPE) cell death [53,54].

In NaIO_3_-induced RD, RPE cells primarily died at 12 h after RD. Subsequently, photoreceptor cells in the ONL died at 24 h after RD, and their degeneration peaked at RD 72 h. Consistent with findings in blue-LED-induced RD (Figure 2), most microglial cells in the ONL and SRS were IBA1 and P2RY12 co-labeled in normal and RD retinas at 24 h after RD induction (Figure 5A–C). At 72 and 120 h after RD, however, IBA1/P2RY12-co-labeled cells significantly decreased compared with normal and 24 h post-RD retinas (*p* < 0.05). Thus, only IBA1-labeled cells were observed (Figure 5D,E,K). In addition, the time course of changes in CD86 and CD206 expression in microglial cells (Figure 5F–J) was similar to that in the blue-LED-induced RD (Figure 3F). In microglial cells, the expression of CD86, an M1 cell marker, gradually increased in the ONL and SRS from 24 to 72 h after RD, whereas CD206, an M2 marker, abruptly increased at 72 h (Figure 5L). Most microglial cells at this time point were IBA1/CD86/CD206-triple-labeled (Figure 5I,L). Real-time PCR of CD206^high^ microglial cells revealed upregulation of phagocytosis- and metabolism-related genes rather than pro- and anti-inflammatory genes (*p* < 0.05, Figure 5M,N). Taken together, these results suggest that innate microglia exhibit common dynamics in acute RD models, namely, a complex expression of CD86 and CD206 and upregulation of phagocytosis-related genes at the peak RD time-point.

## 3. Discussion

Microglial cells are key players in retinal inflammation during RD. To understand the pathogenesis of and establish therapeutic strategies against RD, we need to characterize microglial cells, including their origin, molecular signature, and function. Recent studies using a mouse line, which allowed inducible tracking of microglia, demonstrated that microglial cells in the SRS are recruited from innate retinal microglial cells instead of the extrinsic monocyte-derived macrophages in light-induced RD [42]. In contrast, microglial cells within the inner retina are mainly monocyte-derived macrophages instead of migrating innate retinal microglial cells [55]. Characteristics of the infiltrated microglial cells in the SRS and ONL were revealed using single-cell analysis [42]. However, they did not describe the expression of the M1 and M2 markers, and thus, the applicability of the M1/M2 markers in the RD retina needs to be re-evaluated.

In this study, microglial cells in the normal retina expressed P2RY12 were primarily located in the OPL and IPL, and their processes were horizontally ramified. However, in the early phase (up to 24 h after RD), in both blue LED- and NaIO_3_-induced RD models, microglial cells were detected in the ONL and SRS. Microglial cells in the ONL have a distinct morphology showing vertically elongated processes (Figure 1B,C, Figure 2B,C and Figure 5B,C). In addition, they still expressed P2RY12 (Figure 2B,C and Figure 5B,C). In contrast to an increase in microglial cells in the ONL and SRS, microglial cells in the OPL and IPL abruptly decreased after 12 h (Figure 2H). A summation of these results strongly suggests that activated and migrating microglial cells found in the ONL and SRS were derived from innate microglial cells originally localized in the OPL and IPL. Nonetheless, the total number of microglia increased in the retina during peak RD (Figure 1F–H and Figure 2F–H). Thus, the origin of the increased microglial cells must be investigated. These cells could originate from the proliferation of innate cells or another cell type, such as extrinsic monocyte-derived macrophages.

Expression of P2RY12 in microglial cells within the retina remained until 24 h after RD but disappeared by the RD peak time of 72 h, when numerous activated microglia infiltrated the ONL, including in giant phagocytic form (Figure 2D). Similar findings at the genetic level have been reported in various retinal disease models, such as light-induced RD [42], glaucoma [56], and choroidal neovascularization [57]. Taken together, these results suggest that when innate microglial cells are activated in response to retinal injury, they lose their homeostatic characteristics. This assumption is supported by similar findings for another homeostatic gene, *tmem119*, which is prominently downregulated along with *p2ry12* in light-induced RD [42]. Therefore, homeostatic markers, including P2RY12, may not be useful as innate microglial markers after activation of microglial cells under pathological conditions in the retina.

Microglial cells are commonly classified into pro-inflammatory M1 and anti-inflammatory M2 types [32,33,46,58]. Thus, the transformation of M1 or M2 may be an important strategy for modulating neuroinflammation. However, this M1/M2 classification of microglial cells and its therapeutic effect on RD are still debated since many studies, especially in vivo studies, lack supporting evidence for it [16,17,18,19]. Recent studies recommend avoiding the use of M1 and M2 classification [59]. Thus, the expression of the M1 and M2 markers needs to be re-evaluated in the retinal microglia cells to determine their applicability.

Many studies have investigated microglial cells in RD with a focus on M1/M2 polarization [36,58,60]. However, the spatiotemporal distribution and functional significance of microglial cells expressing M1/M2 markers remain unclear. For instance, while Zhou et al. [36] identified a CD86/CD206-co-expressing microglial cell population using FACS, they did not elucidate its spatiotemporal profile in response to RD and its role in the pathogenesis of RD.

In this study, we clearly demonstrated a spatiotemporal change of the M1/M2 marker in retinal microglial cells followed by progression of RD through our triple-labeling experiments with anti-IBA1 antibody, anti-CD86 antibody (a representative M1 marker), and anti-CD206 antibody (a representative M2 marker). Between 12 h to 24 h after RD induction, in the early phase of RD, most of the activated microglial cells in the RD site were IBA1/CD86-double-labeled, but not CD206-labeled, in the two RD models (Figure 3 and Figure 5). Meanwhile, in the active (or peak) phase of RD, at 72 h, IBA1/CD86/CD206-triple-labeled microglial cells abruptly increased, accounting for most of the microglial cells. This complex expression of the conventional M1 and M2 markers was demonstrated in both blue LED-induced RD and NaIO_3_-induced RD models. These results suggest that a simple M1/M2 classification may not be appropriate for characterizing microglial cells in RD based on their markers. Furthermore, the dynamic spatiotemporal profile of IBA1/CD86/CD206-triple-labeled microglial cells in response to RD extends our understanding of CD86/CD206-co-expressing microglial cells, as they were first introduced in a genetic RD model, the rd1 mice [36].

What is the main role of CD86/CD206-co-expressing microglial cells corresponding to IBA1/CD86/CD206-triple-labeled microglial cells in this study in RD? Despite recent studies using single-cell RNA sequencing to reveal various microglial cell subtypes and their functions, little is known about CD86 and CD206 in microglial cells as these studies mainly focused on functional and origin-related genes of microglial cells [41,42,61]. However, CD86 and CD206 are still used as M1/M2 markers in recent microglial cell studies [62,63], and thus the dominant role of CD86-, CD206-, and CD86/CD206-co-expressing microglial cells needs to be further evaluated. In this study, we sorted microglial cells using anti-CD206 antibody. IBA1/CD206-double-labeled population can reflect the majority of IBA1/CD86/CD206-triple-labeled cells because most (92.8%) IBA1/CD206-double-labeled microglial cells also express CD86 at the peak time of RD. We isolated CD206^+^ microglia and evaluated their function using real-time PCR. Interestingly, neither *Il-6* nor *Il-10* were increased in the CD206^high^ microglial cells compared with that in CD206^low^ microglial cells, which means that IBA1/CD86/CD206-triple-labeled cells may not exhibit primarily pro- and anti-inflammatory function in RD. Meanwhile, expression of *Trem-2*, *Lyz2*, and *Apoe* was significantly increased in the CD206^high^ microglial cells compared with that in CD206^low^ microglial cells (Figure 4D). *Trem-2* is a representative phagocytic gene that encodes TREM-2, a surface protein that recognizes a target and initiates and enhances phagocytosis [52,64,65,66]. *Apoe* is a type of apolipoprotein generally involved in lipid metabolism [67,68]. APOE can activate microglial cells and promote phagocytosis by interacting with TREM-2 [69]. *Lyz2* is the most highly upregulated disease-associated gene in activated microglial cells expressing *Apoe* and could be a marker for active phagocytosis [70]. As mentioned above, CD206 is also a crucial surface marker for phagocytosis [71,72,73]. Taken together, these results suggest that IBA1/CD86/CD206-triple-labeled microglial cells are responsible for phagocytosis in RD rather than inflammation. Our immuno-EM result (Figure 4E) also supports the phagocytic function of microglial cells in the ONL and SRS, as described in previous studies [24,28,42].

To enforce that the dominant role of microglial cells in RD is phagocytosis, we tried to figure out the origin of IL-10, a representative anti-inflammatory M2 cytokine. We performed real-time PCR using the whole retina and found that the expression of *Il-10* increased with no statistical difference at 24 h after LED exposure but significantly increased at 72 h (Appendix A). Since the RT-PCR results showed there is no increase in *Il-10* levels in FACS-sorted CD206 microglial cells, we hypothesized that Müller glial cells may be a source of *Il-10*. Hence, we sorted Müller glial cells using an anti-CD44 antibody, a representative Müller glial marker [74,75] and used the sorted cells to measure *Il-10* levels. CD44 sorted cells showed an increase of the glutamine synthetase (GS), a representative Müller glial marker, and it showed that *Il-10* was greatly enhanced in the Müller glial cells at 72 h (Appendix A). Our results indicate that microglial cells play a major role in phagocytosis in RD, and anti-inflammatory response could be mediated by Müller glial cells. Therefore, microglial cells in RD could not be classified as either M1 or M2.

The advantages of the phagocytic function of microglial cells in RD are still debated. While phagocytosis can aggravate RD by engulfing both apoptotic and non-apoptotic photoreceptors [24], it can also protect photoreceptors by removing cell debris [28]. This conflicting effect of microglial phagocytosis is observed in various CNS diseases. Phagocytosis by microglia promotes neuronal loss in numerous types of brain injuries, while it may have either a beneficial or deleterious role in the degeneration process of brain neurons [76,77]. Therefore, the newly identified dominant microglial subpopulation, CD86/CD206-double-labeled cells, which are mainly involved in phagocytosis, is important to understand the role of phagocytosis in RD. We suggest that modulating the phagocytic function of microglial cells could be a potential target for treating RD. Additionally, we presented the critical time of microglial cell phenotype change during RD by showing the spatiotemporal change of microglial cells. Those CD206 expression and polarization of the microglia could be a critical factor for developing new treatment methods [78]. However, further studies are needed to evaluate the influence of microglial phagocytosis on the progression of RD, such as controlling the expression of the CD206.

In summary, we examined the spatiotemporal distribution of microglial cells in two mouse models of RD. Features of activated microglial cells in RD could be described using two key time points. First, in the early phase of RD, innate microglial cells migrate to the degeneration site (the ONL and SRS) and maintain expression of P2RY12 (homeostatic marker) and express CD86 (M1 marker). Second, when RD peaks and shows prominent photoreceptor cell death, CD206 (M2 marker) abruptly increases, and P2RY12 disappears. Our results demonstrate the consecutive change of the innate microglial cells, P2RY12^+^/CD86^−^/CD206^−^ for normal, P2RY12^+^/CD86^+^/CD206^−^ for early RD, and P2RY12^-^/CD86^+^/CD206^+^ for active RD (Appendix A, schema). We suggest that the population type and role of microglial cells are different in different RD stages. Finally, we described the CD86/CD206-double-labeled microglial population; they expressed phagocytosis-related genes, such as *Trem2*, *Apoe*, and *Lyz2*, at high levels. These results suggest that microglial cells in the active RD retina could not be exactly classified as either M1 or M2, and phagocytosis is the dominant role of microglial cells during active RD.

## 4. Materials and Methods

### 4.1. Animals

In this study, 6-week-old, male BALB/c (n = 50) and C57BL/6J (n = 50) mice were used. The mice were kept in climate-controlled conditions, with a 12 h light/dark cycle. The study protocol was approved by the Institutional Animal Care and Use Committee of the School of Medicine, The Catholic University of Korea (Approval number: CUMS-2017-0241-03, CUMS-2019-0266-06), which acquired AAALAC International full accreditation in 2018.

### 4.2. Animal Models

Blue LED-induced RD and NaIO_3_-induced RD models were used to investigate the changes in microglial cells during RD, as described in detail in our previous studies [23,79,80]. Briefly, BALB/c mice were dark-adapted for 24 h, and then their pupils were dilated with Mydrin P (Santen Pharmaceutical, Osaka, Japan) under dim red-light conditions for 20 min. The mice were then exposed to 1800 lux blue LED (460 ± 10 nm) for 2 h in cages with reflective interiors. Following blue LED exposure, the mice were kept in a dark room for 1 h and moved to a climate-controlled room with a 12 h light-dark cycle.

For the NaIO_3_-induced RD model, 35 mg/kg of NaIO_3_ (Sigma, St. Louis, MO, USA) was injected into the C57BL/6 mice via a single intraperitoneal (IP) injection. Control mice were injected with an equal volume of 0.9% normal saline.

### 4.3. Tissue Preparation

All mice were anesthetized with an intraperitoneal injection of zolazepam (20 mg/kg) and xylazine (7.5 mg/kg). The anterior segments of the eyes were removed, and only the eyecups were fixed in 4% paraformaldehyde for 2.5 h. Then, the eyecups were rinsed in 0.1 M phosphate buffer (PB; pH 7.4), transferred to 30% sucrose in 0.1 M PB, infiltrated overnight, and embedded in O.C.T. compound (Sakura Finetek, Alphen aan den Rijn, The Netherlands) for frozen tissue specimens. Eyecups were cut to 8 μm thickness using cryostat at −25 °C.

### 4.4. Terminal Deoxynucleotidyl Transferase dUTP Nick End Labeling (TUNEL) Assay

TUNEL assays were performed in 8 µm thickness cryosections according to the manufacturer’s protocols using the In Situ Cell Death Detection kit (Roche Biochemicals, Mannheim, Germany) to detect the death of retinal cells. Retinal sections were washed 3 times for 10 min in 0.01 M phosphate-buffered saline (PBS) and then incubated with permeabilization solution (0.1% Triton-100, 0.1% sodium citrate) for 2 min on ice. Subsequently, sections were incubated with terminal deoxynucleotidyl transferase enzyme at 37 °C for 1 h. After washing, cell nuclei were counterstained with 4′,6-diamidino-2′-phenylindole (DAPI; dilution, 1:1000; Invitrogen, Eugene, OR, USA) for 10 min. Cells were visualized using a Zeiss LSM 800 Meta confocal microscope (Carl Zeiss, Oberkochen, Germany).

### 4.5. Immunohistochemistry

Cryosections were washed three times with 0.01 M PBS, then blocked with 10% normal donkey serum in 0.2% Triton X-100 in PBS for 1 h at room temperature. They were then incubated overnight at 4 °C with the following primary antibodies (Table 1). The next day, sections were incubated with secondary antibodies for 2 h at room temperature. Cell nuclei were counterstained with DAPI, as described above. All sections were mounted using mounting media (Vector Laboratories, Burlingame, CA, USA). Images were taken using a Zeiss LSM 800 Meta confocal microscope. Quantitative analysis was performed on five vertical retinal sections from each group. In each retinal section, we manually counted immuno-labeled microglial cells within 700 μm of the optic disc, which is the region where severe RD usually occurs [23,79,80]. This value was represented as the cell number/mm.

### 4.6. Fluorescence-Activated Cell Sorting (FACS)

For FACS analysis, we modified the protocol described in a previous report [41]. Retinas were incubated with 2.5 mL of digestion buffer containing Hank’s balanced salt solution (HBSS; Thermo Fisher Scientific, Waltham, MA, USA), 5% fetal bovine serum (FBS; Capricorn, Palo Alto, CA, USA), 10 mM HEPES (Sigma), 0.7 mg/mL calcium chloride (Sigma), 1.5 mg/mL collagenase A (Roche, Basel, Switzerland), and 0.1 mg/mL DNase I (Roche) at 37 °C for 15 min. The retinas were then gently dissociated into a single-cell suspension with a pipette and filtered through a 70 µm cell strainer. The cells were then centrifuged at 300× *g* for 5 min, resuspended in PBS containing 2% FBS, and counted. For the next step, cells were blocked with 5% normal donkey serum for 15 min at 4 °C. Cells were then incubated with anti-rabbit CX3CR1 and anti-goat CD206 antibodies for 20 min at 4 °C, followed by incubation with Alexa Fluor 488 (anti-rabbit) and Cy3 (anti-goat) secondary antibodies for 15 min at room temperature. The cell suspensions were washed with PBS, filtered through 70 µm cell strainer, and transferred to a FACS tube (Falcon, Reynosa, Mexico) at a concentration of 1 × 10^6^/mL. Cells were sorted using FACSCanto II (BD, Franklin Lakes, NJ, USA), and data analysis was performed using FlowJo software (v10.6.2, BD).

### 4.7. One-Step Real-Time Polymerase Chain Reaction (qRT-PCR)

Total RNA was extracted from the FACS-sorted cell populations using the Cell-Amp TM Direct RNA Prep Kit (TaKaRa, Kyoto, Japan). We then used the One Step TB Green^®^ PrimeScript TM PLUS RT-PCR Kit (TaKaRa) to measure the RNA levels of interleukin 6 (*Il-6*), interleukin 10 (*Il-10*), apolipoprotein E (*Apoe*), lysozyme 2 (*Lyz2*), and triggering receptor expressed on myeloid cells (*Trem2*) in each group. Primer sequences for each gene are listed in Table 2. All data were normalized actin as an endogenous control and analyzed using the delta-delta Ct method.

### 4.8. Immuno-Electron Microscopy (EM)

Retinas from blue LED-induced RD mice were fixed with 4% paraformaldehyde in 0.1 M PB for 2 h and transferred to 2.3 M sucrose in 0.1 M PB for 24 h for dehydration. The retinas were then frozen in liquid nitrogen and cut into semi-thin cryosections (2 μm) at −100 °C using a Leica EM UC7 ultramicrotome equipped with an FC7 cryochamber (Leica, Wetzlar, Germany). The sections were blocked with 10% normal donkey serum in PBS for 1 h at room temperature and then labeled at 4 °C overnight using a 1:100 dilution of CD206 antibody. After washing in PBS, the samples were incubated with a secondary donkey anti-rabbit-peroxidase antibody for 1 h (1:100, Sigma). Tissue sections were washed with PBS, followed by rinsing in 0.05 M TB (Tris-HCL, Biosesang, Incheon, Republic of Korea). The sections were then incubated with DAB solution for a few minutes, washed with 0.1 M PB for 10 min, and post-fixed with 2.5% glutaraldehyde and 1% osmium tetroxide for 30 min. Silver enhancement was performed using the HQ silver enhancement kit (Nanoprobes, Yaphank, NY, USA) for 3 min. Sections were dehydrated in graded alcohol and embedded in Epon 812 (Polysciences, Warrington, PA, USA). Areas of interest, selected under light microscopy, were cut into ultrathin sections (80–90 nm) and observed under an electron microscope (JEM 1010, Tokyo, Japan).

### 4.9. Statistical Analysis

Statistical analysis of the number of microglial cells and gene expression levels was performed using Prism 8.0 software (GraphPad, San Diego, CA, USA). Differences between two groups were analyzed by Student’s *t*-test, and differences between more than two groups were analyzed by one-way ANOVA and Tukey’s multiple comparison test. Statistical significance was set at *p* < 0.05.

## Figures and Tables

**Figure 1 ijms-24-14236-f001:**
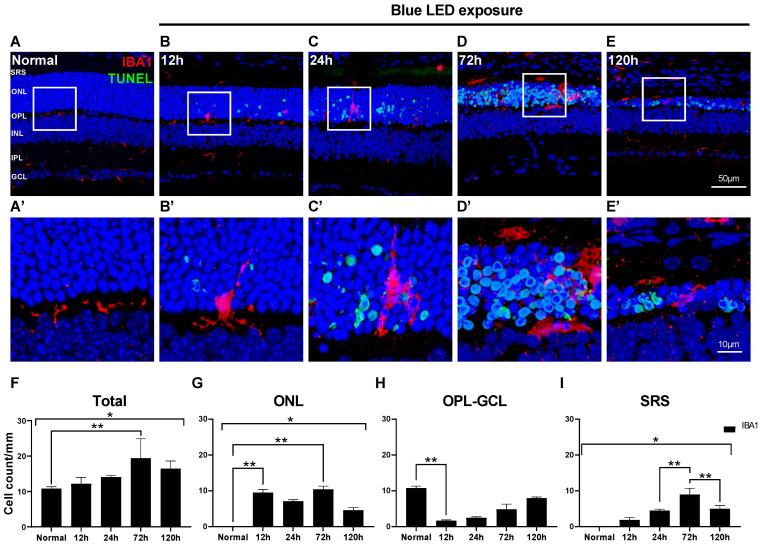
Distribution of microglial cells in blue LED-induced retinal degeneration (RD). Microglial cells and degenerating photoreceptors were labeled with anti-IBA1 antibody (red) and TUNEL (green). (**A**–**E**). Representative images of IBA1/TUNEL-double-labeled vertical sections taken from normal retina (**A**) and RD retinas at 12 (**B**), 24 (**C**), 72 (**D**), and 120 h (**E**). Each boxed area is magnified in (**A’**–**E’**), respectively. Scale bars: 50 μm (**A**–**E**), 10 μm (**A’**–**E’**). (**F**–**I**). Quantitative analyses of the IBA1-labeled microglial cells in RD. The number of the IBA1-labeled microglial cells was counted in the range of 700 μm from the optic disc in retinal vertical sections and presented as cell number per mm in each layer: whole retina (**F**), ONL (**G**), OPL to GCL (**H**), and SRS (**I**) (n = 5). Data are presented as the mean ± S.E.M. * *p* < 0.05, one-way ANOVA with Tukey’s multiple comparison post-hoc test, ** *p* < 0.05, Tukey’s multiple comparison test. GCL, ganglion cell layer; IPL, inner plexiform layer; INL, inner nuclear layer; OPL, outer plexiform layer; ONL, outer nuclear layer; SRS, subretinal space.

**Figure 2 ijms-24-14236-f002:**
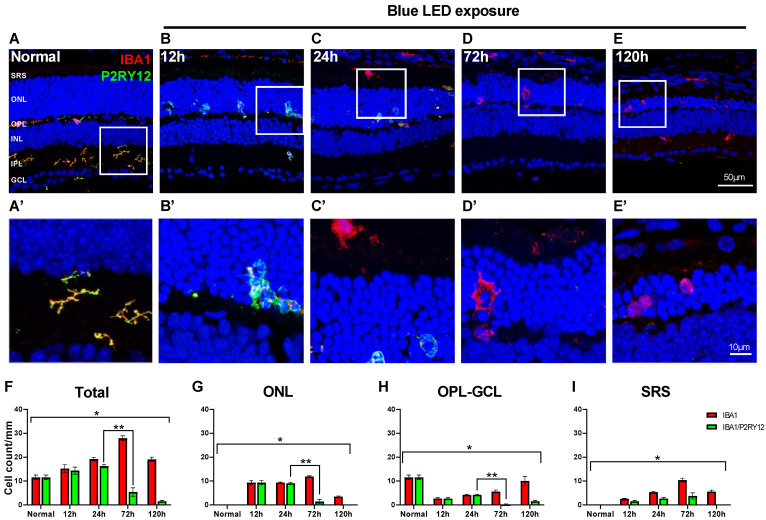
Response of the innate microglial cells in blue LED-induced RD. Anti-IBA1 antibody (red) and anti-P2RY12 antibody (green) were used as markers for pan-microglial cell and innate microglial cell populations, respectively. (**A**–**E**). Representative images of IBA1/P2RY12-double-labeled vertical sections taken from normal retina (**A**) and RD retinas at 12 (**B**), 24 (**C**), 72 (**D**), and 120 h (**E**). Each boxed area is magnified in (**A’**–**E’**), respectively. Scale bars: 50 μm (**A**–**E**), 10 μm (**A’–E’**). (**F**–**I**). Quantitative analyses of the IBA1/P2RY12-double-labeled microglial cells in RD. The cell number was counted in the range of 700 μm from the optic disc in retinal vertical sections and presented as cell number per mm by each layer: whole retina (**F**), ONL (**G**), OPL to GCL (**H**), and SRS (**I**) (n = 5). Data are presented as the mean ± S.E.M. * *p* < 0.05, one-way ANOVA with Tukey’s multiple comparison post-hoc test, ** *p* < 0.05, Tukey’s multiple comparison test. GCL, ganglion cell layer; IPL, inner plexiform layer; INL, inner nuclear layer; OPL, outer plexiform layer; ONL, outer nuclear layer; SRS, subretinal space.

**Figure 3 ijms-24-14236-f003:**
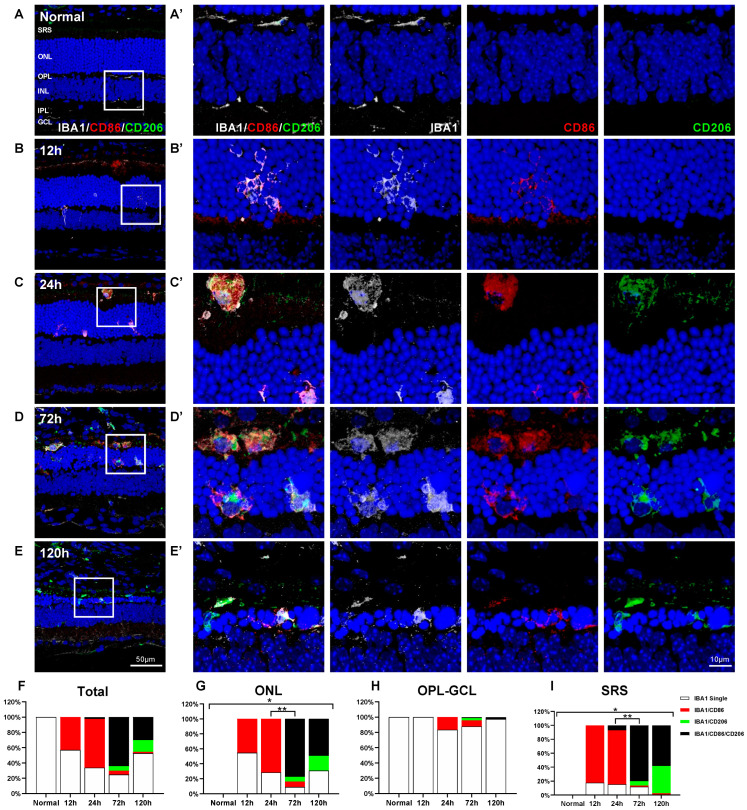
Responses of the M1 and/or M2 microglial cells in blue LED-induced retinal degeneration (RD). Whole population of microglial cells, M1, and M2 cells were labeled with anti-IBA1 antibody (white), anti-CD86 antibody (red), and anti-CD206 antibody (green), respectively. (**A**–**E**). Representative images of IBA1/CD86/CD206-triple-labeled vertical sections taken from normal retina (**A**) and RD retinas at 12 (**B**), 24 (**C**), 72 (**D**), and 120 h (**E**). Each boxed area is magnified in (**A’**–**E’**), and the following three panels show IBA1-, CD86-, and CD206-channels, respectively. The first of them are a merging of three antibodies labeled images. Scale bars: 50 μm (**A**–**E**), 10 μm (**A’**–**E’**). (**F**–**I**). Quantitative analyses of the IBA1 single-labeled cells, IBA1/CD86-double-labeled cells, IBA1/CD206-double-labeled cells, and IBA1/CD86/CD206-triple-labeled cells in RD. Moreover, 100% stacked column chart shows their proportions. The cell number was counted in the range of 700 μm from the optic disc in retinal vertical sections and presented as cell number per mm by each layer: whole retina (**F**), ONL (**G**), OPL to GCL (**H**), and SRS (**I**) (n = 5). Data are presented as the mean ± S.E.M. * *p* < 0.05, one-way ANOVA with Tukey’s multiple comparison post-hoc test, ** *p* < 0.05, IBA1/CD86/CD206-triple-labeled cells number, Tukey’s multiple comparison test. GCL, ganglion cell layer; IPL, inner plexiform layer; INL, inner nuclear layer; OPL, outer plexiform layer; ONL, outer nuclear layer; SRS, subretinal space.

**Figure 4 ijms-24-14236-f004:**
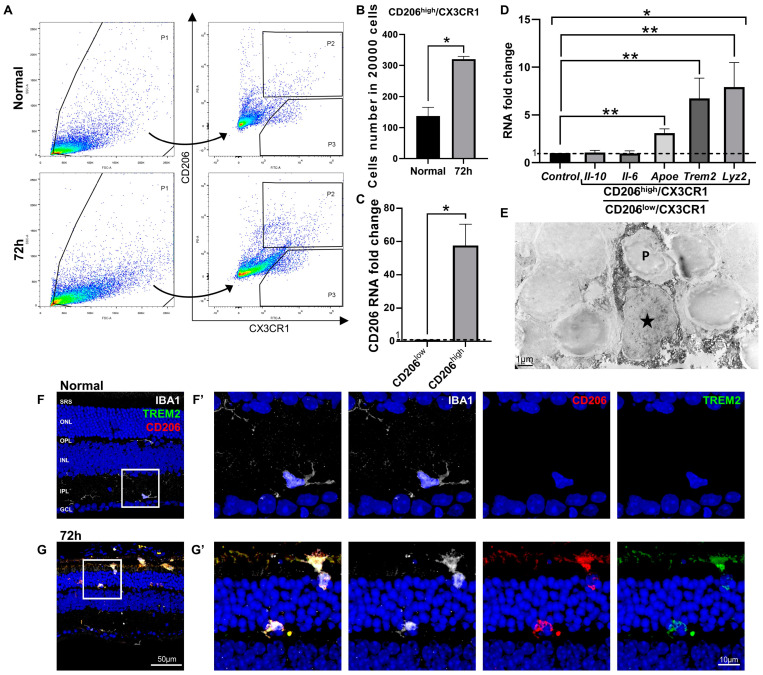
Isolation and characterization of the CD206-labeled (IBA1/CD86/CD206-triple-labeled) microglial cells in blue LED-induced retinal degeneration (RD). (**A**). Microglial cells from the eyecups of the normal and RD 72 h retinas were sorted using FACS with anti-CX3CR1 antibody and anti-CD206 antibody. Cells were sorted into two groups, CD206^high^/CX3CR1 (P2) and CD206^low^/CX3CR1 (P3). (**B**). The number of the CD206^high^/CX3CR1 cells per 20,000 retinal cells in the normal and 72 h retina. (**C**). The RNA expression level of CD206 between CD206^high^/CX3CR1 and CD206^low^/CX3CR1 cells groups (n = 5, * *p* < 0.05, Student’s *t*-test). (**D**). The RNA expression level of the representative pro-inflammatory (*Il-6*), anti-inflammatory (*Il-10*), and phagocytosis-related (*ApoE*, *Trem2*, and *Lyz2*) genes between CD206^high^/CX3CR1 and CD206^low^/CX3CR1 microglial cells (n = 5, * *p* < 0.05, Student’s *t*-test). The fold-change of the RNA expression level was calculated using the delta-delta CT method after qRT-PCR. (**E**). Immuno-electron microscopy (EM) with anti-CD206 antibody taken from the outer nuclear layer (ONL) of the retina 72 h post-RD. A star (★) indicates cell body of a dark DAB-labeled microglial cell (MG), of which the cytoplasm is engulfing a dying photoreceptor (P). Scale bar: 1 μm. (**F**,**G**). Representative images of IBA1/CD206/TREM2-triple-labeled vertical sections taken from normal (**F**) and 72 h post-RD retinas (**G**). Each boxed area was magnified in (**F’**–**G’**). Scale bars: 50 μm (**F**,**G**), 10 μm (**F’**,**G’**). Data are presented as the mean ± S.E.M. * *p* < 0.05, one-way ANOVA with Tukey’s multiple comparison post-hoc test, ** *p* < 0.05, Tukey’s multiple comparison test. GCL, ganglion cell layer; IPL, inner plexiform layer; INL, inner nuclear layer; OPL, outer plexiform layer; ONL, outer nuclear layer; SRS, subretinal space; *Il-10*, interleukin 10 gene; *Il-6*, interleukin 6 gene.

**Figure 5 ijms-24-14236-f005:**
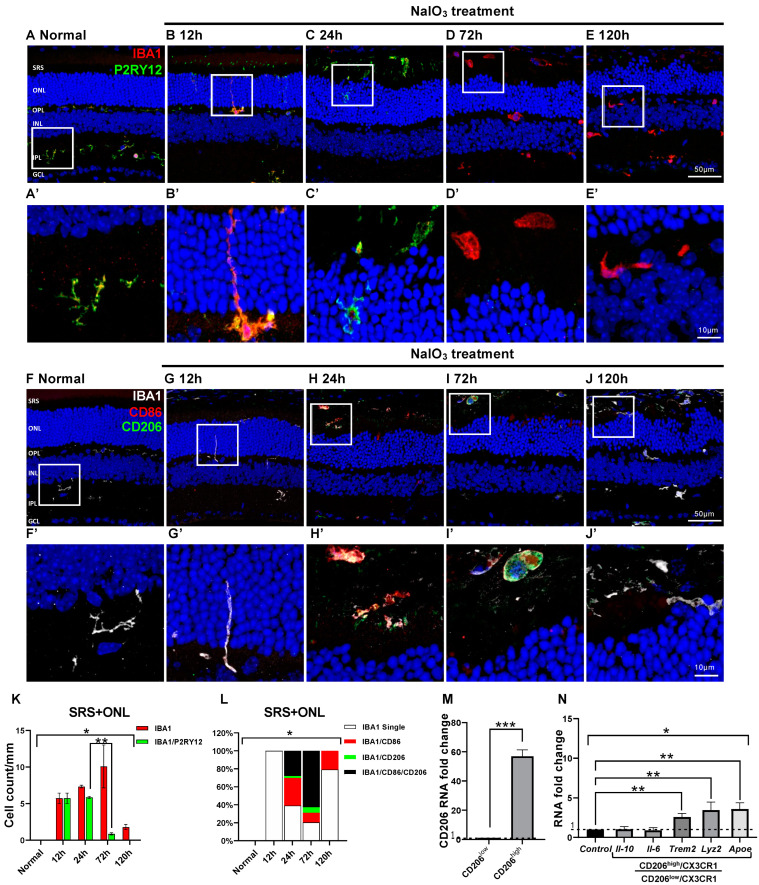
Microglial cell response and characters in the NaIO_3_-induced RD. Anti-IBA1 antibody (red) and anti-P2RY12 antibody (green) were used to label the pan-microglial cells and innate microglial cells. (**A**–**E**). Representative images of IBA1/P2RY12-double-labeled vertical sections taken from normal (**A**) and RD retinas at 12 (**B**), 24 (**C**), 72 (**D**), and 120 h (**E**). Each boxed area is magnified in (**A’**–**E’**), respectively. Quantitative analyses of the IBA1/P2RY12-double-labeled microglial cells in the ONL and SRS showed significant loss of P2RY12 at 72 h (**K**). Representative images of IBA1/CD86/CD206-triple-labeled vertical sections taken from normal retina (**F**) and RD retinas at 12 (**G**), 24 (**H**), 72 (**I**), and 120 h (**J**). Each boxed area is magnified in (**F’**–**J’**), and the following three panels show IBA1-, CD86-, and CD206-channels, respectively. Scale bars: 50 μm (**A**–**J**), 10 μm (**A**’–**J**’). Quantitative analyses of the IBA1/CD86-double-labeled M1, IBA1/CD206-double-labeled M2, and IBA1/CD86/CD206-triple-labeled cells in the ONL and SRS (**L**). Moreover, 100% stacked column chart shows their proportions. The cell number was counted in the range of 700 μm from the optic disc in retinal vertical sections and presented as cell number per mm (n = 5). (**M**). The RNA expression level of the CD206 between CD206^high^/CX3CR1 and CD206^low^/CX3CR1 cells groups (n = 5). (**N**). The RNA expression level of the representative pro-inflammatory (*Il-6*), anti-inflammatory (*Il-10*), and phagocytosis-related (*ApoE*, *Trem2*, and *Lyz2*) genes between CD206^high^/CX3CR1 and CD206^low^/CX3CR1 microglial cells (n = 5). The fold-change of the RNA expression level was calculated using the delta-delta CT method after qRT-PCR. Data are presented as the mean ± S.E.M. * *p* < 0.05, one-way ANOVA with Tukey’s multiple comparison post-hoc test, ** *p* < 0.05, Tukey’s multiple comparison test, *** *p* < 0.05, Student’s *t*-test. GCL, ganglion cell layer; IPL, inner plexiform layer; INL, inner nuclear layer; OPL, outer plexiform layer; ONL, outer nuclear layer; SRS, subretinal space; *Il-10*, interleukin 10 gene; *Il-6*, interleukin 6 gene.

**Table 1 ijms-24-14236-t001:** Primary antibodies.

Target	Species	Catalog	Company	Concentration
anti-IBA1	Rabbit	019-19741	Wako Pure Chemical Industries	1:500
anti-IBA1	Goat	ab5076	Abcam	1:500
anti-CD206	Goat	AF2535	R&D Systems	1:250
anti-P2RY12	Rabbit	AS-55043A	AnaSpec	1:500
anti-CD86	Rat	ab119857	Abcam	1:250
anti-CD44	Rat	103002	BioLegend	1:250
anti-IL-10	Mouse	505002	BioLegend	1:100
anti-TREM2	Sheep	AF1729	R&D Systems	1:50

**Table 2 ijms-24-14236-t002:** PCR Primer sequence information.

Gene	Species	Forward (5′→3′)	Reverse (5′→3′)
*Actin*	Mouse	CAT TGC TGA CAG GAT GCA GAA GG	TGC TGG AAG GTG GAC AGT GAG G
*CD206*	Mouse	TCA TCC CTG TCT CTG TTC AGC	ATG GCA CTT AGA GCG TCC AC
*Apoe*	Mouse	GGG ACA GGG GGA GTC CTA TAA T	TTT GCC ACT CGA GCT GAT CT
*Trem2*	Mouse	GAC CTC TCC ACC AGT TTC TCC	TAC ATG ACA CCC TCA AGG ACT G
*Lyz2*	Mouse	TGA ACG TTG TGA GTT TGC CAG	CAG CAG AGC ACT GCA ATT GAT
*IL-10*	Mouse	CGG GAA GAC AAT AAC TGC ACC C	CGG TTA GCA GTA TGT TGT CCA GC
*IL-6*	Mouse	TAC CAC TTC ACA AGT CGG AGG C	CTG CAA GTG CAT CAT CGT TGT TC

## Data Availability

The datasets used and/or analyzed during the current study are included in this article and available from the corresponding author upon reasonable request.

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
