# Peer review of "A Distinct Microglial Cell Population Expressing Both CD86 and CD206 Constitutes a Dominant Type and Executes Phagocytosis in Two Mouse Models of Retinal Degeneration"

_ijms, 2023, doi:10.3390/ijms241814236_

Round 1

Reviewer 1 Report (New Reviewer)

Major comments:

The authors aim to clarify if the M1/M2 classification reflects the functional diversity of the microglial cells in the degenerating retina. Thereby, the detection of representative M1/M2 markers and related functional genes revealed that the majority of microglia express both CD86 and CD206.  The presented results demonstrate that microglial cells in active retinal degeneration, represented by the blue LED light and sodium iodate models, cannot be classified as M1 or M2 and that the majority of them is involved in phagocytosis rather than inflammation.

The introduction is very short and does not touch the physiological role of microglia in the eye and the issues with distinguishing them from macrophages. As this is also important to understand the role of microglia in the retina, it should be added.

FACS sorting of the triple-positive microglia cells is questionable. In order to characterize the different cell populations, also the CXCR3+/CD86+ have to be similarly analyzed. Following this, the two populations (CXCR3+/CD206+ vs. CXCR3+/CD86+) can then be compered as described. This is also valid for the Without this analysis, the results might be rather without definite proof.

The retinal degeneration induced by sodium iodate (Line 275-277) has to be presented. Retinal damage after i.p. injection of 35 mg/kg in such short time intervals is questionable. Many publications state other kinetics including concentration and time differences between the application modes.

Minor comments:

Line 66-68: Why did the authors not count the number of TUNEL-positive cells in the retina? As without quantification the TUNEL staining doesn’t make sense. This would also be a proof for the induced degeneration.

Line 130–131: Was the mentioned recovery significant? If not, then it would be an overinterpretation of the results.

Line 330: Where are the data from 4h in NaIO3 model?

Figure 6N: These data have not been described in the text and the legend.

Figure 2C: The image magnification should be adjusted to cover both labeled cells completely.

Figure 2E: No co-labeled cells are visible.

Figure 3C: The image magnification should be adjusted to cover all visible labeled cells completely.

Legend Figure 4: The legend describes “MG” as label for the microglia cell, but in the figure there is an “*” shown.

Supplemental Figure S1: In order to follow the argumentation of the authors better, it would be recommended to order the panels as described in the text from A to I.

Author Response

Manuscript ijms-2542079

Response to Reviewers1

Thank you for your kind and important comment on our manuscript. It improves our manuscript.  We carefully read your feedback and tried to make the best answer, and if needed, we performed additional experiments. Please consider our effort to complete this manuscript.

Reviewer 1

Major comments:

Q1. The authors aim to clarify if the M1/M2 classification reflects the functional diversity of the microglial cells in the degenerating retina. Thereby, the detection of representative M1/M2 markers and related functional genes revealed that the majority of microglia express both CD86 and CD206. The presented results demonstrate that microglial cells in active retinal degeneration, represented by the blue LED light and sodium iodate models, cannot be classified as M1 or M2 and that the majority of them is involved in phagocytosis rather than inflammation.

The introduction is very short and does not touch the physiological role of microglia in the eye and the issues with distinguishing them from macrophages. As this is also important to understand the role of microglia in the retina, it should be added.

Response: Thank you for your comments. We agree that our introduction needs more explanation for microglial cells in the eye. We added a more detailed explanation of the microglial function and distinguishing the macrophage in the introduction. Please see the line 44-47, 52-59, 61-65, and 67-68.

Q2. FACS sorting of the triple-positive microglia cells is questionable. In order to characterize the different cell populations, also the CXCR3+/CD86+ have to be similarly analyzed. Following this, the two populations (CXCR3+/CD206+ vs. CXCR3+/CD86+) can then be compered as described. This is also valid for the Without this analysis, the results might be rather without definite proof.

Response: Thank you for your comments. We only tested CD206 sorted cells because almost all microglial cells in the injury site expressed both CD86 and CD206 at 72 h, and CD206 was abruptly increased at 72 h which is an important finding of phase change of the microglial cells. As you pointed out, evaluation of CX3CR1+/CD86+ could be important, thus we compared CX3CR1+/CD86- and CX3CR1+/CD86+ from 72 h RD retina. In the results, CX3CR1+/CD86+ showed an increase of il-6 and phagocytic markers (TREM2, CD206), while il-10 was not changed. We added qPCR results for revision in this response letter. We included the results in Word file.

1) The increase of CD206 and Trem2 in the CD86+ cells indicates that microglial cells in the active RD state generally express both CD86 and CD206, and it is related to phagocytosis. 

2) Then why CD86 sorted cells show an increase of il-6 while CD206 sorted cells did not? That might happen because of CD86 single expression cells, which we presented in Fig. 3. As we described in the manuscript, CD86 was increased in the early stage of RD, and it changed to CD86/CD206 double expression cells at 72 h RD, but still, CD86 single positive cells were also minimally remained. CD86 is a representative pro-inflammatory marker, thus it could induce il-6 expression, rather than CD206 sorted cells.

Q3. The retinal degeneration induced by sodium iodate (Line 275-277) has to be presented. Retinal damage after i.p. injection of 35 mg/kg in such short time intervals is questionable. Many publications state other kinetics including concentration and time differences between the application modes.

Response: NaIO3-induced photoreceptor degeneration model is frequently used for studying RD. The time course of the RD by the NaIO3 depends on the injection root and injection amount of the NaIO3. Generally, NaIO3 showed typical degeneration processes by systemic injection root, single IP or IV injections. It showed RPE degeneration from 12 h ~24 h, and it showed prominent photoreceptor cell death around 72 h. We added the previous studies which showed similar characteristics of the NaIO3 model below the answer. Of course, we evaluated our NaIO3-induced RD model by TUNEL, but it is the same as the previous reports. Thus, we just showed the microglial cell changes.

  1. Sodium Iodate-Induced Mouse Model of Age-Related Macular Degeneration Displayed Altered Expression Patterns of Sumoylation Enzymes E1, E2 and E3, Current Molecular Medicine, 2018.
  2. Sodium Iodate-Induced Degeneration Results in Local Complement Changes and Inflammatory Processes in Murine Retina. Int J Mol Sci, 2021.
    c. Morphologic characteristics of retinal degeneration induced by sodium iodate in mice. Current Eye Research, 2002.
  3. Functional Recovery of Retina After Sodium Iodate Injection in Mice. Vision Research, 1996.
  4. Retinal pigment epithelial cell necroptosis in response to sodium iodate. Cell death discorvery, 2016

Minor comments:

Q1. Line 66-68: Why did the authors not count the number of TUNEL-positive cells in the retina? As without quantification the TUNEL staining doesn’t make sense. This would also be a proof for the induced degeneration.

Response: We apologize that we skipped counting the TUNEL-positive photoreceptor cells. We already reported the TUNEL response of our blue-LED induced RD model in our previous studies, cited in the manuscripts with reference numbers 23, 79, and 80, which showed initiation of the TUNEL at 24 h, reaching the peak at 72 h.

And, this study pointed to the microglial cell change, not the photoreceptor cell death. Based on the TUNEL response in our previous reports and the focus on the microglia, we did not include the TUNEL-positive cell number in the results, because it is enough to understand the microglial cell change. Please consider our previous reports and the purpose of this study.

Q2. Line 330: Where are the data from 4h in NaIO3 model?

Response: We apologize for this careless mistake. It should be 24 h, we revised in the manuscript. Please see the line 340.

Q3. Figure 6N: These data have not been described in the text and the legend.

Response: We apologize for this careless mistake. We thought that you pointed Figure 5N. We changed “Fig. 5M” to “Fig. 5M-N” in lines 289-290. And, we already described Fig. 5N in the figure legend.

Q4, Figure 2C: The image magnification should be adjusted to cover both labeled cells completely.

Response: Thank you for your comments. In Fig. 2C, the green-labeled cell in the ONL, which is located at the corner, is a both-labeled cell. As you pointed we changed the magnification of the A’-E’, to cover a wider area and include the whole morphology of the cells in Fig. 2C. Please see Fig. 2.

Q5. Figure 2E: No co-labeled cells are visible.

Response: Thank you for pointing out the significance. Images in Fig.2 are captured in the core region of the injury site, a distance from 250~400 μm from the optic nerve. However, IBA1/P2RY12 co-labeled cells at 120 h were mainly detected in the IPL around the border of the blue-LED injury area, thus it was not detected at the core region of the injury site in Fig.2. As we described in the method, immnuo-labeled cells in the 700 μm distance from optic nerve were manually counted in the tile-scan image, thus we could not insert the image contained co-labeled cells. Instead, we added the image in this Word file.

Q6. Figure 3C: The image magnification should be adjusted to cover all visible labeled cells completely.

Response: All visible labeled cells were presented in the first column of each figure. We intended to magnify the specific cells in lower magnification images, thus we could not contain every cell in the inset view in Fig. 3C. But we contained important findings in Fig.3C, which present IBA1/CD206/CD86 triple positive cells in the SRS, and IBA1/CD86 double-positive cells in the ONL. It could explain the main results in the manuscript.

Q7. Legend Figure 4: The legend describes “MG” as label for the microglia cell, but in the figure there is an “*” shown.

Response: We apologize for our mistake. We added “MG” in Figure 4.

Q8. Supplemental Figure S1: In order to follow the argumentation of the authors better, it would be recommended to order the panels as described in the text from A to I.

Response: Thank you for your comments. As you pointed out, we changed the order of the figure to match the description in the text. Please see the Fig. S1.

Reviewer 2 Report (New Reviewer)

General comments

This article is deemed to be of overall good quality. In their work, the authors challenge the conventional classification of mouse retinal microglial cells (RMGs hereafter) into M1 and M2 phenotypes during activation in the blue LED-damaged retina. RMGs are characterized in regards to their functional significance, spatiotemporal evolution and profile distribution during retinal degeneration (RD). The finding is reported that a mixed (M1/M2), triple-labeled cell subtype becomes the predominant RMG subtype along RD progression, and their phagocytic (rather than pro- or anti-inflammatory) function is emphasized. In the light of this and other results the conclusion is drawn that a simple M1/M2 classification may not be appropriate for characterizing RMG populations in RD. Besides, coherent findings are obtained in a second mouse model, namely the NaIO3‒-induced RD model. Experiments are elegantly conducted, the results are sound and the conclusions drawn in the Discussion are convincing. Nonetheless, a good number of sites are here pointed out where redaction must be amended/improved or a clarification needs to be made before the article is deemed publishable in IJMS.

Minor points

Please avoid the use of the article the in the vast majority of instances preceding ‘(retinal) microglial cells’, ‘CD86’ or ‘CD206’. For instance, in lines 52, 93, 368 and 374 (RMGs); lines 150, 188 and 252 (CD86); and line 189 (CD206). Conversely, the must be added preceding ‘dominant role’ in line 409.

In the sentence ‘... in microglial cells and their movement during RD progression.’ (line 58), I believe the authors mean ‘... their movement across retinal layers during RD progression’. Please amend if agreed.

IBA1 is presented as a selective marker of retinal pan-microglial cells. This includes both inactive and activated microglia? Please clarify (line 67).

Please replace ‘entire/whole retinal layer’, which is confusing, by ‘entire/whole retina (lines 80, 100, 145 and 203).

Please clarify what IBA1/P2RY12 colabeled RMGs represent (line 122).

I believe the term ‘inactivated’ must be replaced by inactive (line 158). Do the authors agree?

What is the role of CX3CR1+ cells in the normal and degenerating retina? (lines 188-189). Is CX3CR1 a specific molecular marker of a particular RMG subtype? Please specify if so in p. 7 (line 213) and/or Discussion.

 ‘Interestingly, neither Il-6 nor Il-10 were increased...’ (lines 393-394). Please amend as indicated. Likewise, replace ‘and’ by nor in line 396.

‘This conflicting effect... is observed in various CNS diseases’ (lines 423-424). Please give examples.

CD44 was increased in the end-feet of the Müller glial cells in 72 h. Please rewrite (line 555).

Predominant RMGs were IBA1+ P2RY12+ at earlier RD times (24 h) and (after decrease) IBA1+ CD86+ CD206+ at later RD times (72-120 h), in both mouse models. Can P2RY12+ cells be found at late RD times? Do they express (or are expected to express) IBA1, CD86 and/or CD206 markers? Please discuss or show experimental results if available.

Figure 1 legend needs some rewriting. For instance, ‘by each layer’ (line 100) need be replaced by ‘in each layer’, and ‘by the mean’ (line 101) need be replaced by ‘as the mean’.

Figure 4E legend: There is no MG (line 258) in the EM picture, and ‒conversely‒ the asterisk meaning is not indicated in the legend.

Figure S1 title and legend (lines 554-558): GS (glutamine synthetase) is not mentioned in the ms. text. Is this a marker of which cell type(s)/subtype(s)? Please indicate in the text.

Finally, this paper would greatly benefit from adding a simple schema to the Discussion illustrating changes experienced by RMGs during RD regarding phases, migration across retinal layers, molecular markers, MG populations, etc. (described in lines 433-446).

Minor English revision is required throughout the ms.

Author Response

Reviewer 2

General comments

This article is deemed to be of overall good quality. In their work, the authors challenge the conventional classification of mouse retinal microglial cells (RMGs hereafter) into M1 and M2 phenotypes during activation in the blue LED-damaged retina. RMGs are characterized in regards to their functional significance, spatiotemporal evolution and profile distribution during retinal degeneration (RD). The finding is reported that a mixed (M1/M2), triple-labeled cell subtype becomes the predominant RMG subtype along RD progression, and their phagocytic (rather than pro- or anti-inflammatory) function is emphasized. In the light of this and other results the conclusion is drawn that a simple M1/M2 classification may not be appropriate for characterizing RMG populations in RD. Besides, coherent findings are obtained in a second mouse model, namely the NaIO3‒-induced RD model. Experiments are elegantly conducted, the results are sound and the conclusions drawn in the Discussion are convincing. Nonetheless, a good number of sites are here pointed out where redaction must be amended/improved or a clarification needs to be made before the article is deemed publishable in IJMS.

Minor points

Q1. Please avoid the use of the article the in the vast majority of instances preceding ‘(retinal) microglial cells’, ‘CD86’ or ‘CD206’. For instance, in lines 52, 93, 368 and 374 (RMGs); lines 150, 188 and 252 (CD86); and line 189 (CD206). Conversely, the must be added preceding ‘dominant role’ in line 409.

Response: Thank you for your point. As you recommended, we delete ‘the’ in line 103, 160, 198, 263, 378, 384, 419.

Q2. In the sentence ‘... in microglial cells and their movement during RD progression.’ (line 58), I believe the authors mean ‘... their movement across retinal layers during RD progression’. Please amend if agreed.

Response: Thank you for your comments. We revised it as you pointed out. Please see the lines 67-68.

Q3. IBA1 is presented as a selective marker of retinal pan-microglial cells. This includes both inactive and activated microglia? Please clarify (line 67).

Response: Thank you for your comments. As you pointed out, we used IBA1 to label pan-microglial cells. We add it to the manuscript. Please see line 77.

Q4. Please replace ‘entire/whole retinal layer’, which is confusing, by ‘entire/whole retina’ (lines 80, 100, 145 and 203).

Response: Thank you for pointing this out. We have uniformly used “whole retina” in our text. See the lines 90, 110, 155, 213

Q5. Please clarify what IBA1/P2RY12 co-labeled RMGs represent (line 122).

Response: We apologize for not enough description for IBA1/P2RY12. As we described in line 114-115, P2RY12 is a representative innate microglia marker, which is used to discriminate microglia in neural tissue and recruited from system circulation. IBA1/P2RY12 means innate microglia, not from systemic circulation. We added it in line 110-111.

Q6. I believe the term ‘inactivated’ must be replaced by inactive (line 158). Do the authors agree?

Response: Thank you for your comment. We agree with your comment. We revised it accordingly. Please see line 168.

Q7. What is the role of CX3CR1+ cells in the normal and degenerating retina? (lines 188-189). Is CX3CR1 a specific molecular marker of a particular RMG subtype? Please specify if so in p. 7 (line 213) and/or Discussion.

Response: Thank you for your comment. To improve the understanding of the manuscript, we added the explanation for CX3CR1. CX3CR1 is a surface marker for the whole microglia. Thus, it is frequently used to tag microglia in the retina or brain. Please see the lines 223-224.

Q8. ‘Interestingly, neither Il-6 nor Il-10 were increased...’ (lines 393-394). Please amend as indicated. Likewise, replace ‘and’ by nor in line 396.

Response: We apologize for our mistake. We revised it as you pointed out. Please see the line 393.

Q9. ‘This conflicting effect... is observed in various CNS diseases’ (lines 423-424). Please give examples.

Response: Thank you for your comment. The examples are included in the references, with the reference numbers 76 and 77. We added an explanation of it briefly in the manuscript. Please see the line 434-437.

Q10. CD44 was increased in the end-feet of the Müller glial cells in 72 h. Please rewrite (line 555).

Response: We are sorry that we could not exactly understand your comments. We presented the increase of CD44 in the outer limiting membrane, consisting of the end-feet of the Müller glial cells. For a better description, we added white arrows in the normal image. Please see the Figure S1.

Q11. Predominant RMGs were IBA1+ P2RY12+ at earlier RD times (24 h) and (after decrease) IBA1+ CD86+ CD206+ at later RD times (72-120 h), in both mouse models. Can P2RY12+ cells be found at late RD times? Do they express (or are expected to express) IBA1, CD86 and/or CD206 markers? Please discuss or show experimental results if available.

Response: Normally, microglial cells are located in the OPL and IPL with the expression of the P2RY12. However, after injury, microglial cells rapidly migrated to the injury site (ONL~SRS), thus the number of microglial cells in the IPL decreased, and migrated microglial cells in the injury site showed a loss of the P2RY12. At 120 h, the injury started to terminate, thus microglia cells in the IPL were increased to return to the normal state, and microglial cells in the ONL were decreased. And, at 120 h, microglial cells in the IPL showed expression of the P2RY12, while microglial cells in the ONL still have no P2RY12. At 120 h, microglial cells in the IPL do not express CD86 and CD206, and microglial cells in the ONL still express CD86 and CD206.

Q12. Figure 1 legend needs some rewriting. For instance, ‘by each layer’ (line 100) need be replaced by ‘in each layer’, and ‘by the mean’ (line 101) need be replaced by ‘as the mean’.

Response: Thank you for pointing this out. Revised accordingly. Please see the line 110.

Q13. Figure 4E legend: There is no MG (line 258) in the EM picture, and ‒conversely‒ the asterisk meaning is not indicated in the legend.

Response: Thank you for your comments. We added “MG” in Figure 4.

Q14. Figure S1 title and legend (lines 554-558): GS (glutamine synthetase) is not mentioned in the ms. text. Is this a marker of which cell type(s)/subtype(s)? Please indicate in the text.

Response: Thank you for pointing this out. We revised it. Please see line 569, we used the term GS-labeled mϋller glial cells.

Q15. Finally, this paper would greatly benefit from adding a simple schema to the Discussion illustrating changes experienced by RMGs during RD regarding phases, migration across retinal layers, molecular markers, MG populations, etc. (described in lines 433-446).

Response: Thank you for your comments. As you recommended, we added a schema to visualize and summarize our results. Please see the Fig. S2.

Reviewer 3 Report (New Reviewer)

Zhang et al. present a scientific paper describing the scientific planning and aim, targeting whether the M1/M2 classification exactly reflects the functional classification of the microglial cells in the retina, remains debatable. As written in the introduction, this study is meant to support the growing evidence suggesting that microglial cells cannot be simply classified as M1 and M2, thus lacks originality. However, the introduction, methodology used to solve the raised questions in the introduction, experimental plans and their executions, data generation and their interpretations, results, and discussion are well elaborated in this scientific manuscript. To make it more interesting for readers, authors must describe clearly how these findings will help patients.

Author Response

To Reviewer 3
Thank you for your comment on our manuscripts. We carefully read your feedback and try to make an best answer. As you pointed out, the clinical implication of the microglia is important. But it has limitations to expand it. Nevertheless, it has a possibility for clinical use, because specific markers for subtypes of the microglial cells indicate that modulatory method for it.

  1. Zhang et al. present a scientific paper describing the scientific planning and aim, targeting whether the M1/M2 classification exactly reflects the functional classification of the microglial cells in the retina, remains debatable. As written in the introduction, this study is meant to support the growing evidence suggesting that microglial cells cannot be simply classified as M1 and M2, thus lacks originality. However, the introduction, methodology used to solve the raised questions in the introduction, experimental plans and their executions, data generation and their interpretations, results, and discussion are well elaborated in this scientific manuscript. To make it more interesting for readers, authors must describe clearly how these findings will help patients.

Response. Benefit of the inhibiting microglia is debated issues because its role in neurodegenerative disease is controversial. Thus understanding the microglia is important to regulate neuroinflammation. In this study, we demonstrated the spatiotemporal change of the microglial cells followed by the progress of RD, and we revealed that phagocytosis is the main role of the microglia in the active RD state.

 We expect that those findings could help in modulating the microglial cells to helpful pathways. However, the advantage of the phagocytic microglial cells for patients needs more study, because its profit during RD is still controversial. Thus we could not describe definitely whether it helps patients or not. We already described it in the discussion. Nevertheless, CD206 is still a promising target for a new treatment method for RD, thus we added a short sentence about it with reference. Please consider the background, and see the lines 442-444.

Round 2

Reviewer 1 Report (New Reviewer)

All requests have been satisfactorily answered.

Legend S1: Delete "." before "after", change to "Müller" and write "FACS" instead of FACs

Author Response

Manuscript ijms-2542079

Response to Reviewers 1

We apologize that we could not perfectly follow your recommendation in the first revision. Thank you for your kind and careful recommendation, and we revised the manuscript as you pointed out.

Reviewer 1

Minor comments:

Q1. Legend S1: Delete "." before "after", change to "Müller" and write "FACS" instead of FACs

Response: We apologize for these careless mistakes. They have all been corrected in the revised version of our manuscript. Please see the Legend S1.

Reviewer 2 Report (New Reviewer)

General comments

I must insist on some (many!) of my previous queries before the ms. is deemed acceptable, since I see they have not been appropriately fulfilled by the authors.

Minor points

Q1. There are still (a total of 20!) instances in which the article the precedes “microglial cells”. The authors must ensure that the is removed in all of these instances, this occurring (at least, as far as I have seen) in lines 17, 51, 60-65, 126, 166, 199-200, 254, 270-271, 397, 400, 418, 436, 444, 456 and 460.

Q3. “... to label the pan-microglial cells...” in line 77 must be replaced by “... to label pan-microglial cells...”.

Q5. The redaction “... because IBA1/P2RY12-co-labeled microglial cells indicates innate microglial cells.” In lines 120-121 does not sound appropriate. Please better replace it by “Cells co-labeled with both antibodies represented innate microglial cells”. Do the authors agree?

Q7. “... which is common marker...” in lines 224-225 must be replaced by “... which is a common marker...”.

Q9. Please amend lines 435-438 so as to read: Phagocytosis by microglia promotes neuronal loss in numerous types of brain injuries, while it may have either a beneficial or deleterious role in the degeneration process of brain neurons.

Q10. “CD44 was increased in the end-feet of the GS-labeled müller glial cells in 72h...” in line 570 must be replaced by “... CD44 expression was increased in the end-feet of GS-labeled Müller glial cells at 72 h...”

Q13. Figure 4E legend. I must insist: What does the asterisk represent?!

Q14. As far as I know, GS is a specific Müller glial cell marker. If the authors agree with me, please state it somewhere in the text.

Q15. The schema in (new) Fig. S2 looks and seems satisfactory to this referee. Still, a legend must be added to the figure, and P2ry12 must be replaced by P2RY12.

A further, minor English revision of the ms. would be still convenient.

Author Response

Manuscript ijms-2542079

Response to Reviewers2

We apologize that we could not perfectly follow your recommendation in the first revision. We followed your comments as possible as we could in the second revision and tried to best. Thank you for your kind and careful recommendation, and it improves the quality of our manuscript.

Reviewer 2

General comments

I must insist on some (many!) of my previous queries before the ms. is deemed acceptable, since I see they have not been appropriately fulfilled by the authors.

Minor points

Q1. There are still (a total of 20!) instances in which the article ‘the’ precedes “microglial cells”. The authors must ensure that the is removed in all of these instances, this occurring (at least, as far as I have seen) in lines 17, 51, 60-65, 126, 166, 199-200, 254, 270-271, 397, 400, 418, 436, 444, 456 and 460.

Response: We apologize that we could not revise these incorrect use of ‘the’ as you mentioned. We carefully read again, and tried to revise it. Please see the line 17, 51, 60-65, 125, 165, 198-199, 253, 269-270, 332, 396, 399, 417, 456 and 460.

Q3. “... to label the pan-microglial cells...” in line 77 must be replaced by “... to label pan-microglial cells...”.

Response: We apologize that we used ‘the’ incorrectly. As you mentioned, we revised it. Please see the line 77.

Q5. The redaction “... because IBA1/P2RY12-co-labeled microglial cells indicates innate microglial cells.” In lines 120-121 does not sound appropriate. Please better replace it by “Cells co-labeled with both antibodies represented innate microglial cells”. Do the authors agree?

Response: Thank you for your comments. We agree with your recommendation. We revised it accordingly. Please see the line 119-120.

Q7. “... which is common marker...” in lines 224-225 must be replaced by “... which is a common marker...”.

Response: We apologize for our incorrect grammar. We revised it accordingly. Please see the line 223.

Q9. Please amend lines 435-438 so as to read: Phagocytosis by microglia promotes neuronal loss in numerous types of brain injuries, while it may have either a beneficial or deleterious role in the degeneration process of brain neurons.

Response: We agree with your recommendation. We revised it accordingly. Please see the line 435-438.

Q10. “CD44 was increased in the end-feet of the GS-labeled müller glial cells in 72h...” in line 570 must be replaced by “... CD44 expression was increased in the end-feet of GS-labeled Müller glial cells at 72 h...”

Response: We agree with your recommendation. We revised it accordingly. Please see the line 570.

Q13. Figure 4E legend. I must insist: What does the asterisk represent?!

Response: The star (not the asterisk) in Figure 4E legend represents cell body of a dark DAB-labeled microglial cell. We have revised it in the line 269.

Q14. As far as I know, GS is a specific Müller glial cell marker. If the authors agree with me, please state it somewhere in the text.

Response: As you pointed out, we added information about GS, and revised sentences in discussion. Please see the line 426-428.

Q15. The schema in (new) Fig. S2 looks and seems satisfactory to this referee. Still, a legend must be added to the figure, and P2ry12 must be replaced by P2RY12.

Response: Thank you for your comments. As you recommended, we added legend in lines 578-580.

This manuscript is a resubmission of an earlier submission. The following is a list of the peer review reports and author responses from that submission.

Round 1

Reviewer 1 Report

In this work the authors analyze the retinal microglia activation in induced paradigms of retinal degeneration and they conclude that thus activation does not follow the classical M1 or M2 polarization. This conclusion does not contribute to the scientific advance of this field, as this dichotomy, although useful as a preliminary classification, is not longer supported in neurodegenerative processes. The authors understimates the results from single-cell transcriptomics as this is the current approach that has finally demonstrated beyond doubt that the biology of microglia is complex and its activation mostly depends on the type of stimulation, showing a large heterogeneity that cannot be simplified as M1 or M2. Considering this, it is not surprising the main result obtained by the authors, that most of the microglia are Cd86+/Cd206+ without a defined M1 or M2 phenotype, as already reported (see PMID:28928639). This report largely diminishes the novelty of the present study.

As a minor point, IL10 must be named in capital letters following protein nomenclature.

Author Response

Reviewer 1

In this work the authors analyze the retinal microglia activation in induced paradigms of retinal degeneration and they conclude that thus activation does not follow the classical M1 or M2 polarization. This conclusion does not contribute to the scientific advance of this field, as this dichotomy, although useful as a preliminary classification, is not longer supported in neurodegenerative processes. The authors understimates the results from single-cell transcriptomics as this is the current approach that has finally demonstrated beyond doubt that the biology of microglia is complex and its activation mostly depends on the type of stimulation, showing a large heterogeneity that cannot be simplified as M1 or M2. Considering this, it is not surprising the main result obtained by the authors, that most of the microglia are Cd86+/Cd206+ without a defined M1 or M2 phenotype, as already reported (see PMID:28928639). This report largely diminishes the novelty of the present study.

Ans.) We sincerely appreciate your valuable comments and insights. We completely agree with your point of view. It has been previously reported in a study (PMID:28928639, Microglia Polarization with M1/M2 Phenotype Changes in rd1 Mouse Model of Retinal Degeneration) that CD86/CD206 double-positive microglial cells exist, as you have mentioned. However, we do acknowledge that the previous report has certain limitations, such as lacking comprehensive information on their temporal change from advent to disappearance in RD state and their spatial distribution in a lesion-specific manner, as well as their possible function. Fortunately, our results fill in these gaps and extend the knowledge on this topic. We have provided spatiotemporal changes and phagocytosis-related gene expression in the CD86/CD206 microglial cells in active RD state, which confirms their identity and extends the study to two other RD models.

Furthermore, we have carefully considered your comment that we underestimated the results from single-cell transcriptomics. We recognize that single-cell transcriptomics is a powerful tool and has already identified various complex subpopulations and functions. However, we would like to point out that transcriptomics data contains a vast dataset, which may not include all important targets depending on the researcher's interest. For instance, CD206 and CD86 are crucial targets of microglial cells, but they were not included in past studies using transcriptomic results of the microglia from the degenerative retina. We have highlighted this point in our manuscript (please refer to lines 382-386).

As a minor point, IL10 must be named in capital letters following protein nomenclature.

Ans.) We apologize for our typo. We revised it in the manuscript.

Reviewer 2 Report

Abstract

-        Open abbreviations when mentioned first time as well in the main text, and il should also be IL

-        - conclusion in abstract needs more convince. As well secreted inflammatory cytokines and other cytokiens e.g. IL-8 and MCP-1 should measure. At least open little bit background (in abstract or in the other part of the manuscript) why measured only cytokines IL-6 and IL-10. As well the phenotype of M1 and M2 should open little bit if compared to t CD86/CD206-double-labeled microglial cells, which differences there is.

-        there as well should refer to some positive or negatives findings if those cells are more as phagocytotic than inflammatory

-        introduce shortly all measured results in abstract

-        is the phagocytic phenotype beneficial or harmful, give some conclude as well related to that and example.

-        introduce shortly all results in abstract e.g. TUNEL assay came first time at the results part and microglia location results etc.

Introduction

-        End of the introduction should be one paragraph where is concluded as in general the results of the present study and some general conclusion related to it with some references.

Results

-        Figure 1 Check statistics and add all statistical significant results. e.g. for all timepoints and indicate if statistical significant or not. Now it is confusing. And show which group are compared to each other. Compare all timepoints at least to the normal condition and show all statistics.

-        As well Figure 2 F-I, which group are compared to which one and indicate all significant and not significant results and compared groups.  

-        Figure 3 if second picture is merged of labeled pictures mark it as.

-        How measured cytokine were selected? And why only two cytokines were measured? Probably conclusion needs more detection of different inflammatory markers. Are those microglia spesific or why selected?

-        Figure 4 D shown RNA fold of change is confusing? Should present more clearly. For which those all are compared???

-        FACS results need to open little bit.

-        Figure 5 N is as well confusing and need to modify or open little bit groups for which all measured things belongs.

 Figure legends. Please add abbreviations end of the legends.

Discussion

-        lines 386-387, open cited references of previous studies then it is possible to compare those to the present study.

-        lines 393-395, The origin and background of IL-10 should come as well in the results part. It is important notification.

Materials and Methods

-        In table 1 there could as well be dilution buffer detailes and incubation time with condition. Otherwise table is really nice summary of antibodies.

Supplementary data should refer already in the results part related to some other similar kind results. Now it comes in discussion.

-References included as double. Please, modify.

Author Response

Reviewer 2

We appreciate for the considerable time and effort that you have invested in reviewing our manuscript. The feedback was highly valuable and reasonable, contributing significantly to the enhancement of our paper. We have meticulously addressed your comments point-by-point and incorporated the majority of your suggestions. The revised version of our manuscript clearly indicates the changes made in response to the feedback, which are highlighted in red in main text.

Abstract

We apologize for not being able to provide detailed results in the abstract due to the word limit specified in the IJMS guideline for authors, which restricts the maximum word count to 200. Although we made an effort to include as much information as possible, we were constrained by the word limitation. We have taken into account your comments in our revision, but we still need to balance it with the word limit. Thank you for your understanding.

- Open abbreviations when mentioned first time as well in the main text, and il should also be IL

Ans.) We apologize for any mistakes in our previous version of the abstract and have revised it accordingly. While we were unable to fully expand on all abbreviations due to the word limit, we have provided detailed explanations in the main text of the manuscript.

- Conclusion in abstract needs more convince. As well secreted inflammatory cytokines and other cytokiens e.g. IL-8 and MCP-1 should measure. At least open little bit background (in abstract or in the other part of the manuscript) why measured only cytokines IL-6 and IL-10. As well the phenotype of M1 and M2 should open little bit if compared to CD86/CD206-double-labeled microglial cells, which differences there is.

Ans.) Thank you for your valuable comments. We did consider IL-8; however, we did not include it in this study. Our main objective was to investigate the characteristics of microglial cells in three ways. First, we aimed to determine whether the M1/M2 classification is applicable to the retina. Second, we wanted to track the changes in microglial cell status as RD progresses. Third, we aimed to identify the major population of microglial cells and their expected function during the peak time of RD. We classified microglial cell functions into three categories: pro-inflammation, anti-inflammation, and phagocytosis. To determine the type and role of microglial cells, we used representative functional genes such as IL-6, IL-10, CD206, and TREM2, which have been well established for each function. While IL-8 may be related to phagocytic function and MCP-1 is related to migration and recruitment, they are less helpful than IL-6 and IL-10 in evaluating the categorization of microglial function as pro- or anti-inflammation or phagocytosis.

- There as well should refer to some positive or negatives findings if those cells are more as phagocytic than inflammatory

Ans.) We agree with your point. We showed multiple evidence of the phagocytic dependent role of the majority microglial cells in active RD state and presented them in our results. However, due to word limitation, we could not include every information in our results. As you pointed out, we revised our abstract to reinforce the negative and positive findings of the microglial functions we evaluated. Please see the revised abstract.

- introduce shortly all measured results in abstract

Ans.) As mentioned earlier, due to the word limit, we could not include all our results in the abstract. However, we have made revisions to better highlight our findings, as per your suggestion. Please see the revised abstract.

- is the phagocytic phenotype beneficial or harmful, give some conclude as well related to that and example.

Ans.) As you pointed out, we also considered whether the phagocytic phenotype is beneficial or not. Thus we already discussed it at the end of the discussion. We apologize that we could not include it in the abstract because of the word limit, and it is not our experimental results. Our conclusion in the discussion contained it. So, we added about it more detail in discussion. See the lines 416-427 in Discussion.

- introduce shortly all results in the abstract e.g. TUNEL assay came first time at the results part and microglia location results etc.

Ans.) Following your suggestion, we have added the results in Introduction. Please see the revised abstract.

Introduction

- End of the introduction should be one paragraph where is concluded as in general the results of the present study and some general conclusion related to it with some references.

Ans.) Following your comment, we revised the end of the introduction and added one paragraph. Please see the lines 60-66 in Introduction.

Results

- Figure 1 Check statistics and add all statistical significant results. e.g. for all timepoints and indicate if statistical significant or not. Now it is confusing. And show which group are compared to each other. Compare all timepoints at least to the normal condition and show all statistics.

Ans.) We apologize for our inappropriate presentation in Figure 1. We have revised it as you pointed. The importance of the Fig.1 is that microglial cells were only detected in the OPL and IPL in the normal retina, but microglial cells in the OPL and IPL abruptly migrated to the ONL and SRS from 12 h to 72 h after injury, especially peak at 72 h. Thus, the population of the microglial cells in IPL is decreased. Using the ANOVA, we presented the existence of the statistical differences among all time point and measured the difference between each time point by multiple comparison test. See the revised Figures 1F - I and the figure legend.

- As well Figure 2 F-I, which group are compared to which one and indicate all significant and not significant results and compared groups.  

Ans.) We apologize for our inappropriate presentation in Figure 2. We have revised it as you pointed. In Figure 2, the homeostatic marker (p2ry12) of the microglial cells maintains until 24 h, and significantly decreases at 72 h, which means that microglial cells change their characteristics when the injury is peak. Thus, we pointed abrupt decrease of the IBA1/P2RY12 double-positive cells at 72 h. See the revised Figure 2F - I and the figure legend.

- Figure 3 if second picture is merged of labeled pictures mark it as.

Ans.) Thank you for your comments. We added a mark of the merged labeling in Fig. 3. See the revised Figure 3.

- How measured cytokine were selected? And why only two cytokines were measured? Probably conclusion needs more detection of different inflammatory markers. Are those microglia specific or why selected?

Ans.) IL-6 and IL-10 are commonly used markers to evaluate M1/M2 polarization, with CD86 and CD206 being the respective markers. This has been discussed in various review papers, two of which we have cited in our manuscript (1. Role of Microglial M1/M2 Polarization in Relapse and Remission of Psychiatric Disorders and Diseases. Pharmaceuticals (Basel). 2014 Nov 25;7(12):1028-48. doi: 10.3390/ph7121028. / 2. Targeting Microglial Activation States as a Therapeutic Avenue in Parkinson’s Disease. Front. Aging Neurosci. 08 June 2017). For phagocytosis, we used Lyz2, Apoe and Trem2, which are commonly known as phagocytic genes, and have been explained in our manuscript.

- Figure 4 D shown RNA fold of change is confusing? Should present more clearly. For which those all are compared???

Ans.) We apologize to make you confused. The expression of Il-6, Il-10, Apoe, Trem2, and Lyz2 are compared between CD206high and CD206low. RNA fold change means a relative change of the gene expression level between two groups. Thus, control means gene level of the CD206low/CD86 microglial cells which is set in baseline "1”. See the revised Figure 4 and the figure legend.

- FACS results need to open little bit.

Ans.) We think that provide all FACs results in Figures 4A - D and fully explain purpose of this FACS experiment and results in main text and figure legend. Please read the main text again and see the Figure 4A - D.

- Figure 5 N is as well confusing and need to modify or open little bit groups for which all measured things belongs.

Ans.) We apologize to make you confused. We think it parallels with the above comment about Figure 4D. The expression of Il-6, Il-10, Apoe, Trem2, and Lyz2 are compared between CD206high and CD206low. RNA fold change means a relative change of the gene expression level between two groups. Thus, control means gene level of the CD206low/CD86 microglial cells which is set in baseline "1”. See the revised Figure 5 and the figure legend.

- Figure legends. Please add abbreviations end of the legends.

Ans.) We apologize that we missed abbreviations in the figure legends. We added them in all figure legends. Please see the revised figure legends.

Discussion

- lines 386-387, open cited references of previous studies then it is possible to compare those to the present study.

Ans.) We agree with you that we would better describe more about reference. First, we apologize that reference 69 is not applicable in this point, thus we delete it. And, we revised and described about other reference you pointed, at the end of the discussion. See the lines 416-427 in Discussion.

- lines 393-395, The origin and background of IL-10 should come as well in the results part. It is an important notification.

Ans.) Thank you for noticing the importance of our discussion. We also considered including the IL-10 expression in the Muller glial cells in the results section. However, we thought that the experiments from Muller glial cells could undermine the unity of the results, because we focused on the microglia. Thus, even though it is important, we added it to the supplementary figure to enhance the fluency of our manuscript. Please consider why we place it in the supplementary figure.

Materials and Methods

- In table 1 there could as well be dilution buffer detailes and incubation time with condition. Otherwise table is really nice summary of antibodies.

Ans.) Thank you for this suggestion. However, dilution buffer, incubation time and condition have been already informed in the method text. Please see the lines 483-493.

- Supplementary data should refer already in the results part related to some other similar kind of results. Now it comes in the discussion.

Ans.) This comment is similar to previous one in Discussion. As we responded above, we thought that supplementary data might be better placed in the discussion than results, for the fluency of the manuscript. Please consider it.

- References included as double. Please, modify.

Ans.) We apologize for this careless mistake. Thank you for your reminder. Revised accordingly.

Reviewer 3 Report

 In the manuscript submitted by Zhang et. al. titled as ‘A distinct microglial cell population expressing both CD86 and CD206 constitutes a dominant type and executes phagocytosis in two mouse models of retinal degeneration’, the authors characterized the distribution pattern of microglia during RD, and identified a unique type of microglia which co-expresses CD86, CD206 and other phagocytotic markers, Trem2, Apoe, Lyz2, but no detectable P2RY12 were observed in the late stage of RD. Their results suggest M1/M2 classification is not appropriate for the microglia in RD as other degeneration diseases like AD.

 Overall, this study and findings are of significance to provide more knowledge of the microglia profiles during retinal degeneration. The paper itself is well written and organized with clear logic. Still, I have some major and minor comments.

Major comments:

1. There have been single cell RNAseq profiling data (reference 42), could the author check their dataset to see whether a sub cluster of microglia were identified with the expression of CD86 and CD206.

2. Please use the Fig 1H and 2F as examples to clarify the comparisons with significant differences in Fig. 1F-I, Fig. 2G-I, Fig. 3G &I, 4D, 5K-L&N. They look confusing even referring to the text and figure legend.

Minor comments:

1. Please italicize the gene names in Fig. 4D and 5N.

2. The authors have the “habit” to use ‘anti-’ in front of the cell markers. Such as, Line 63,labeled with anti-IBA1’; Line 90, ‘were labeled with anti-IBA1 (red) and TUNEL (green)’, Line134-15, ‘Anti-IBA1 (red) and anti-P2RY12 (green) were used as markers’, Line 151-153, ‘triple-labeling experiments with anti-IBA1 and two representative M1 and M2 markers, anti-CD86 [43,44] and anti-CD206 [10, 43], respectively’, and others. they are suggested to remove ‘anti-’. Others like ‘Anti-CX3CR1 and anti-CD206 were used’ in Line 206, ‘antibody’ was suggested to be added. In general, the authors are suggested to check whether they mean the markers or the antibodies.

Author Response

Reviewer 3

- In the manuscript submitted by Zhang et. al. titled as ‘A distinct microglial cell population expressing both CD86 and CD206 constitutes a dominant type and executes phagocytosis in two mouse models of retinal degeneration’, the authors characterized the distribution pattern of microglia during RD, and identified a unique type of microglia which co-expresses CD86, CD206 and other phagocytotic markers, Trem2, Apoe, Lyz2, but no detectable P2RY12 were observed in the late stage of RD. Their results suggest M1/M2 classification is not appropriate for the microglia in RD as other degeneration diseases like AD.

 Overall, this study and findings are of significance to provide more knowledge of the microglia profiles during retinal degeneration. The paper itself is well written and organized with clear logic. Still, I have some major and minor comments.

Ans.) We appreciate for the considerable time and effort that you have invested in reviewing our manuscript. The feedback was highly valuable and reasonable, contributing significantly to the enhancement of our paper. We have meticulously addressed your comments point-by-point and incorporated the majority of your suggestions. The revised version of our manuscript clearly indicates the changes made in response to the feedback, which are highlighted in red in main text.

Major comments:

- There have been single cell RNAseq profiling data (reference 42), could the author check their dataset to see whether a sub cluster of microglia were identified with the expression of CD86 and CD206.

Ans.) In the reference 42, which showed RNA-Seq results, they mainly focused on the homeostatic gene, origin of the microglial cells, and metabolic state. It did not include CD86 and CD206.

- Please use the Fig 1H and 2F as examples to clarify the comparisons with significant differences in Fig. 1F-I, Fig. 2G-I, Fig. 3G &I, 4D, 5K-L&N. They look confusing even referring to the text and figure legend.

Ans.) We are sorry to make you confused. We revised the presentation for statistical differences in the figures, as you pointed.

Minor comments:

- Please italicize the gene names in Fig. 4D and 5N.

Ans.) We apologize for our miss writing. We revised it accordingly.

- The authors have the “habit” to use ‘anti-’ in front of the cell markers. Such as, Line 63, ‘labeled with anti-IBA1’; Line 90, ‘were labeled with anti-IBA1 (red) and TUNEL (green)’, Line134-15, ‘Anti-IBA1 (red) and anti-P2RY12 (green) were used as markers’, Line 151-153, ‘triple-labeling experiments with anti-IBA1 and two representative M1 and M2 markers, anti-CD86 [43,44] and anti-CD206 [10, 43], respectively’, and others. they are suggested to remove ‘anti-’. Others like ‘Anti-CX3CR1 and anti-CD206 were used’ in Line 206, ‘antibody’ was suggested to be added. In general, the authors are suggested to check whether they mean the markers or the antibodies.

Ans.) We apologize that we make you confused. As you pointed, we remove ‘anti’ for marker, and we added ‘antibody’ for antibody throughout the whole main text.

Round 2

Reviewer 1 Report

After careful reading of authors' response, I am still not convinced

Author Response

The 2nd revision

After careful reading of authors' response, I am still not convinced.

Ans.) We sincerely apologize if our study did not effectively convey the intended message. Our purpose extends beyond simply demonstrating the limitations of the M1/M2 classification in retinal degeneration models. In this manuscript, we aim to comprehensively present the spatiotemporal dynamics of microglia during RD progression, explore the effective use of M1/M2 markers, and elucidate the expected role of the predominant type of microglia in the active RD state. Our findings offer unique insights that differentiate our study from previous research and transcriptomic results. We were fully committed to providing detailed explanations to ensure a clear understanding of the study's intent and rephrase some parts (in red type) in Discussion. Please see lines 364-369, and 377-392. Lastly, we would greatly appreciate the opportunity to address any questions or concerns you may have.

1) Differences with PMID 28928639
As per your observation, the existence of CD86/CD206 double-positive microglia was indeed described in a previous report, the PMID: 28928639. We have acknowledged this pioneering work and appropriately cited it in our manuscript. However, we also acknowledge that the previous report has some limitations, such as being performed using one RD mutant mouse model, lack of direct identification of double-labeled CD86 and CD206, lack of observation of CD86 and CD206 expression in the retinal vertical section during RD, and not providing gene expression profiles from FACs sorted cells.

We used two different mouse RD models in our study, which provided a distinctive stage of RD from 12 to 72 hours after LED exposure and NaIO3 injection. We were able to discriminate the change of the microglial cells depending on the RD state in these models and showed that the time course of the location and characteristics of the microglial cells followed by RD. Our study revealed detailed spatiotemporal profiles of microglial cells with their characteristics, which are important to understand the role of microglial cells in RD pathogenesis and have not been previously reported, not even in the previous study (PMID 28928639). We also identified CD86 and CD206 double-positive microglial cells and demonstrated at least part of their role in RD.

Therefore, we believe that our study has a novelty compared to previous studies. First, our study confirmed and extended the existence of CD86 and CD206 double-positive microglial cells introduced by the previous study in other RD models. Second, we demonstrated their spatiotemporal profiles in response to RD. Lastly, we suggested their main role in RD. We hope this information addresses your concerns and clarifies the contributions of our study.

2) Continuous use of the M1/M2 marker and its importance

As you mentioned, many researchers, including us, agree that microglia cannot be simply categorized as M1/M2. However, if you search for papers that have studied microglial populations in the retina over the past year, there are still more than 20 papers that use the M1/M2 concept. This means that many researchers are still using this classification.

However, it is clear that the application of M1/M2 markers should be cautious because the classification of M1/M2 is incomplete. Therefore, accurate information and use of M1/M2 markers in retinal degeneration models is needed. In addition, M1/M2 markers themselves are useful for understanding microglial function independent of M1/M2 classification. For example, CD206 has a phagocytic role and therefore cannot be restricted to use as an M2 anti-inflammatory cell marker.

The purpose of this paper was not to simply evaluate the M1/M2 classification of the retina, but to suggest the proper use of CD86 and CD206 and caveats to their use based on our findings.

3) Compensate Single-cell transcriptomics

As you point out, the complex heterogeneity of microglia has been revealed by single-cell transcriptomics. While we acknowledge that single-cell transcriptomics is an irreplaceably powerful method for identifying cell types and functions, it has limitations as primers are chosen based on the researcher's interest and not all genes are included in each cell. For example, the papers by Saban's group that we referenced (40, 42), which are the most widely cited single-cell transcriptome studies of microglia, did not include microglial subtypes expressing M1 and M2 markers. However, this does not diminish the importance of CD86 and CD206 in microglial cell studies. Therefore, further investigation is necessary to understand the role of CD86-, CD206-, and CD86/CD206-co-expressing microglial cells. Our study contributes to this research area by shedding light on the spatiotemporal distribution and function of IBA1/CD86/CD206-triple-labeled microglial cells in RD.
